# HDAC6 inhibition reverses axonal transport defects in motor neurons derived from FUS-ALS patients

Wenting Guo[1,2], Maximilian Naujock[3,4], Laura Fumagalli[1,2], Tijs Vandoorne[1,2], Pieter Baatsen[5], Ruben Boon[6], Laura Ordovás [6,7,8], Abdulsamie Patel[6], Marc Welters[6], Thomas Vanwelden[6], Natasja Geens[1,2], Tine Tricot[6], Veronick Benoy[1,2], Jolien Steyaert[1,2], Cynthia Lefebvre-Omar[9], Werend Boesmans [10], Matthew Jarpe[11], Jared Sterneckert[12], Florian Wegner[3], Susanne Petri[3], Delphine Bohl[9], Pieter Vanden Berghe [10], Wim Robberecht[1,13], Philip Van Damme [1,2,13], Catherine Verfaillie [6] & Ludo Van Den Bosch [1,2]

Amyotrophic lateral sclerosis (ALS) is a rapidly progressive neurodegenerative disorder due to selective loss of motor neurons (MNs). Mutations in the fused in sarcoma (FUS) gene can cause both juvenile and late onset ALS. We generated and characterized induced pluripotent stem cells (iPSCs) from ALS patients with different FUS mutations, as well as from healthy controls. Patient-derived MNs show typical cytoplasmic FUS pathology, hypoexcitability, as well as progressive axonal transport defects. Axonal transport defects are rescued by CRISPR/Cas9-mediated genetic correction of the FUS mutation in patient-derived iPSCs. Moreover, these defects are reproduced by expressing mutant FUS in human embryonic stem cells (hESCs), whereas knockdown of endogenous FUS has no effect, confirming that these pathological changes are mutant FUS dependent. Pharmacological inhibition as well as genetic silencing of histone deacetylase 6 (HDAC6) increase α-tubulin acetylation, endoplasmic reticulum (ER)–mitochondrial overlay, and restore the axonal transport defects in patient-derived MNs.

---

[1] KU Leuven-Department of Neurosciences, Experimental Neurology and Leuven Institute for Neuroscience and Disease (LIND), Leuven 3000, Belgium. [2] VIB, Center for Brain and Disease Research, Laboratory of Neurobiology, Leuven 3000, Belgium. [3] Department of Neurology, Hannover Medical School, Hannover 30625, Germany. [4] CNS Research Department, Boehringer Ingelheim Pharma GmbH & Co. KG, Biberach an der Riß 88400, Germany. [5] VIB Bio Imaging Core and VIB-KU Leuven, Center for Brain and Disease Research, Leuven 3000, Belgium. [6] KU Leuven-Stem Cell Institute (SCIL), Leuven 3000, Belgium. [7] Aragon I+D Foundation (ARAID), Zaragoza 50018, Spain. [8] Biomedical Signal Interpretation and Computational Simulation (BSICoS) Group, Aragon Institute of Engineering Research, IIS Aragón, University of Zaragoza, Zaragoza 50018, Spain. [9] Inserm U 1127, CNRS UMR 7225, Sorbonne Universités, UPMC Univ Paris 6, Institut du Cerveau et de la Moelle épinière (ICM), Hôpital Pitié-Salpêtrière, Paris 75013, France. [10] Lab for Enteric NeuroScience, TARGID, KU Leuven, Leuven 3000, Belgium. [11] Acetylon Pharmaceuticals Inc., Boston, MA 02210, USA. [12] Center for Regenerative Therapies Dresden (CRTD), Technische Universität Dresden, Dresden 01069, Germany. [13] University Hospitals Leuven, Department of Neurology, Leuven 3000, Belgium. Catherine Verfaillie and Ludo Van Den Bosch contributed equally to this work. Correspondence and requests for materials should be addressed to C.V. (email: Catherine.Verfaillie@med.kuleuven.be) or to L.V.D.B. (email: Ludo.Vandenbosch@kuleuven.vib.be)

Amyotrophic lateral sclerosis (ALS) is the most common degenerative disorder of motor neurons (MNs) in adults and is characterized by the selective death of both upper and lower MNs. This wasting of MNs leads to progressive paralysis and death of the patient due to respiratory failure usually within 2 to 5 years after symptom onset[1]. In most cases, ALS is a sporadic disease although ~10% of patients have a clear family history. Mutations in the superoxide dismutase 1 (SOD1), the TAR DNA-binding protein (TARDBP), the fused in sarcoma (FUS), and the chromosome 9 open reading frame 72 (C9orf72) genes are the most prevalent ones[1]. No effective treatment is available for ALS. Riluzole, an FDA-approved drug which has anti-excitotoxic properties, prolongs life by only a few months[2]. Although intensive basic research has been done based on different ALS animal models, none of the discovered strategies has been successfully translated into a therapy[3]. As a consequence, more reliable ALS models are urgently needed. MNs differentiated from patient-derived-induced pluripotent stem cells (iPSCs) provide a new opportunity to model human disease starting from patient material and avoiding effects of species differences and/or overexpression. It can mimic the developmental process in vitro, which could contribute to understand basic disease mechanisms, especially early pathological changes that are potentially amenable to drug screening[4].

FUS was first identified as an oncogene and was reported as an ALS-causing gene in 2009[5–7]. It contains a glycine-rich region, an RNA recognition motif and a nuclear localization signal (NLS). Many point mutations in FUS have been discovered since then and a large number of these are situated in the C-terminal NLS region[8]. FUS can also cause a rare and very aggressive juvenile onset ALS[5, 9]. FUS functions as a DNA/RNA-binding protein and is involved in multiple aspects of DNA/RNA metabolism[8]. The most significant pathological change in post mortem tissue is the cytoplasmic mislocalization of FUS. In iPSC models, cytoplasmic mislocalization of mutant FUS was reported by three independent groups in their patient-derived MNs[10–12]. Cytoplasmic FUS aggregates were observed by Liu et al.[10] for the P525L mutation. In addition, electrophysiological changes have been frequently observed in ALS patient-derived MNs[11, 13–15]. Both hyperexcitability and hypoexcitability have been proposed as pathophysiological defects in these models[11, 13–15]. A recent study based on iPSC-derived MNs carrying mutations in C9orf72 or in TARDBP suggested a switch from hyperexcitability to hypoexcitability, which could result in MN death[12]. For FUS, intrinsic membrane hyperexcitability was presented by Wainger et al.[14], whereas hypoexcitability (characterized by reduced repetitive and spontaneous action potentials, lower synaptic activity and lower intracellular Na+/K+ ratios) was reported by Naujock et al.[13]

There is no clear explanation why MNs selectively degenerate in ALS. One hypothesis is based on the observation that MNs characterized by long axons degenerate first and that altered functions of the most distal sites occurs at initial stages of disease[16]. In this dying-back theory, MNs lose their function at the distal axon and retract back to the MN soma. The longest and largest axons with the highest metabolic demand seem to be the most vulnerable ones, which suggest that defects in axonal transport could be involved in this neurodegenerative process[17]. Most of the energy comes from mitochondria transported to the distal site where they are most needed[17]. In addition, multiple cargos such as proteins, mRNAs, lipids, and organelles are mostly synthesized in the cell body and are transported to the distal part of the axon to maintain their function[18]. About 5–20% of the mitochondria are in very close proximity to the endoplasmic reticulum (ER), which seems to have an important role in many neurodegenerative diseases, including ALS[19, 20]. This region is called the mitochondria-associated ER membrane (MAM) and is linked to intracellular trafficking of mitochondria and ER, Ca$^{2+}$ and phospholipid exchange, energy metabolism, mitochondrial biogenesis, ER stress responses, and autophagy[19, 20]. MAMs were reported to be regulated by TDP-43, which is the protein encoded by TARDBP and by FUS in transfected cells and in mouse models through activation of glycogen synthase kinase-3β (GSK-3β)[21, 22].

We previously discovered that histone deacetylase 6 (HDAC6) inhibitors could rescue axonal transport defects in dorsal root ganglion (DRG) neurons from a transgenic mouse model of the axonal form of Charcot–Marie–Tooth disease (CMT2)[23]. In contrast to the other family members of the HDAC family that mainly deacetylate histones in the nucleus, HDAC6 is localized in the cytoplasm and is the major deacetylating enzyme of α-tubulin[24]. Acetylation of α-tubulin is important for the binding of molecular motor proteins to the microtubules[25, 26]. The importance of HDAC6 in ALS is illustrated by the fact that genetic deletion of HDAC6 significantly slowed disease progression and prolonged survival of the mutant SOD1$^{G93A}$ mouse model[27]. FUS and TDP-43 have also been reported to regulate HDAC6 expression[28–30]. As a consequence, one of our aims was to investigate the therapeutic potential of HDAC6 inhibition.

To further investigate the pathological mechanism and to identify potential therapeutic strategies, we generated iPSCs from fibroblasts of ALS patients carrying different FUS mutations, as well as from family members without mutations. In addition, we generated isogenic control lines using the CRISPR-Cas9 technology and overexpressed wildtype and mutant FUS in human embryonic stem cells (hESCs). Highly pure MN cultures were obtained by differentiating these iPSCs or hESCs. The cells expressing mutant FUS showed cytoplasmic accumulation of FUS, hypoexcitability, and axonal transport defects. We could successfully rescue the axonal transport defects by treating the MNs with HDAC6 inhibitors and by silencing HDAC6 using antisense oligonucleotides (ASOs). Our results indicate that axonal transport could have an important role in ALS pathology, and that HDAC6 inhibition could become a promising therapeutic strategy for FUS-induced ALS.

## Results

**Generation and characterization of iPSCs and MNs**. To investigate the effect of mutations in FUS on MNs, we generated iPSCs from fibroblasts of ALS patients with FUS mutations and controls. Fibroblasts from four different individuals with point mutations in FUS were used. Three carried a heterozygous R521H mutation, whereas another one carried a heterozygous P525L mutation (Supplementary Table 1). For the P525L mutation, we generated multiple lines to exclude possible heterogeneity between different iPSC lines of the same patient. Both R521H and P525L point mutations are localized in the NLS region of the FUS protein. P525L is an aggressive mutation causing rare early onset ALS, and also our patient developed the disease at an early age (Supplementary Table 1)[31]. In addition, we established two control iPSC lines (healthy control1 and healthy control2) from two healthy family members of one of our patients. We also included another two published healthy controls (healthy control3 and healthy control4) for patch clamp measurements[13]. Reprogramming of fibroblasts was performed using Sendai viral vectors expressing 4 reprogramming factors (Klf4, Oct3/4, Sox2, and cMyc). This led to an integration-free conversion of fibroblasts into iPSCs. The absence of Sendai virus was confirmed by qPCR analysis (Supplementary Fig. 1A). The reprogramming efficiency of mutant and control fibroblasts was the same. All iPSC lines expressed pluripotent stem cell markers as determined by semi-quantitative PCR for Oct4, Nanog, Sox2, Rex1 (Fig. 1a) and by immunocytochemical analysis for SSEA4, TRA1-60,

OCT4, NANOG (Fig. 1b). The pluripotency was also confirmed by the analysis of in vivo teratoma formation, which contained the three germ layers (Fig. 1d). All iPSC lines contained the patient-specific *FUS* mutation (Fig. 1c). Furthermore, we established one isogenic control line for the 2/2 patient with the R521H mutation using the CRISPR-Cas9 technology. The genome-edited iPSC line retained the pluripotency markers (Supplementary Fig. 2A) and showed markers of the three germ layers after the embryonic body formation analysis (Supplementary Fig. 2B). The origin of the iPSC lines from the specific

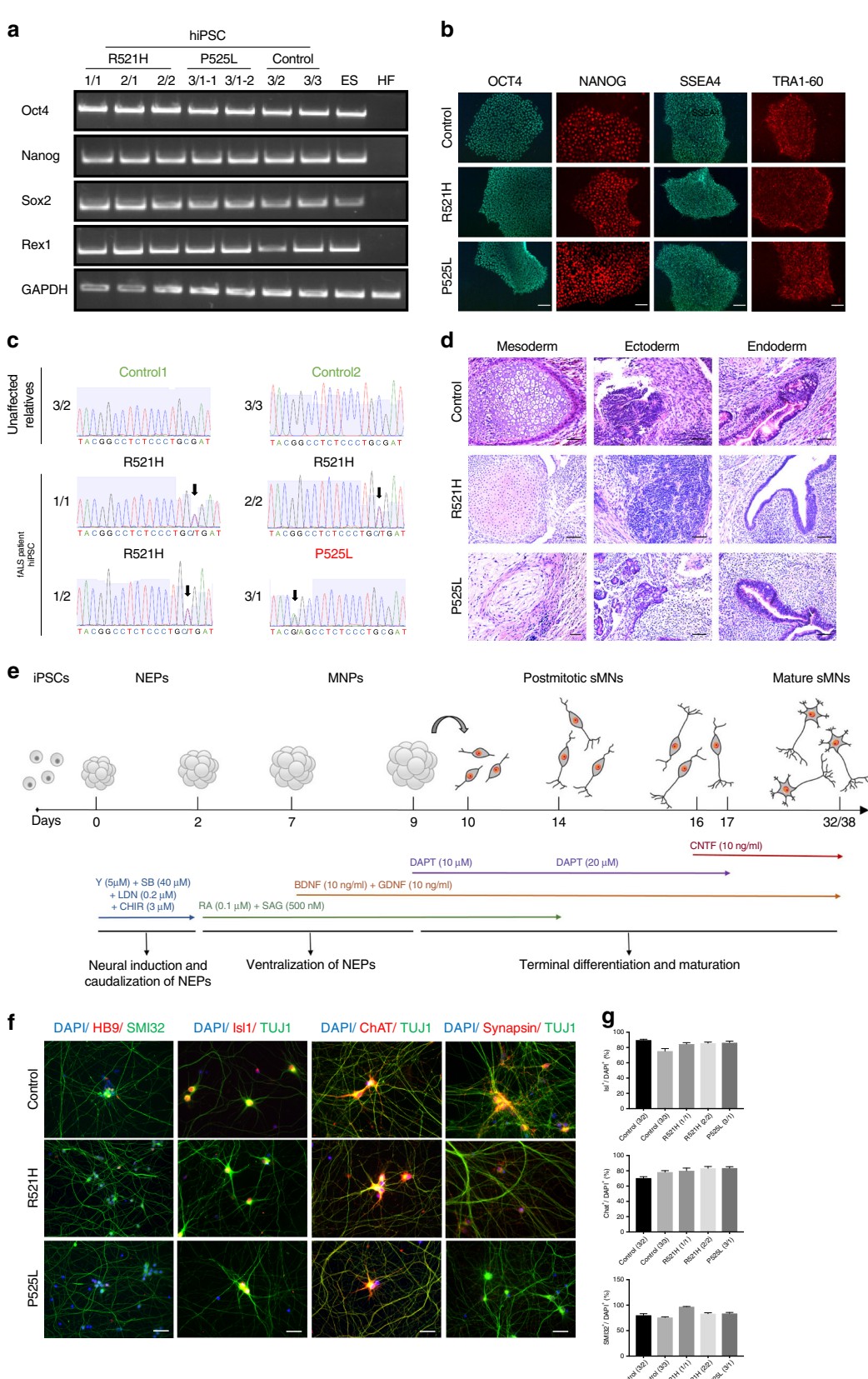

fibroblasts was confirmed by the single nucleotide polymorphism (SNP) analysis (Supplementary Table 2).

To differentiate iPSCs into MNs, we used a slightly modified protocol originally described by Maury et al.[32] (Fig. 1e). We used SMAD inhibition and Wnt activation to obtain neuroepithelial stem cells (NEPs). Subsequently, caudalization was induced by retinoic acid (RA), followed by induction of ventralization using a combination of RA, and smoothened agonist (SAG). On day 9, DAPT, a γ-secretase inhibitor, was added to enhance the conversion of spinal MN progenitors into post-mitotic spinal MNs. Immature MNs obtained at day 17 were maintained in culture with ciliary neurotrophic factor (CNTF) to allow further maturation. We found no differences in MN differentiation capability in the different mutant lines, in the isogenic genetically corrected R521R line, and in the different controls (Fig. 1f; Supplementary Fig. 2C). A high number of cells stained positive for the typical markers including HB9, Isl1, ChAT, Synapsin1, and SMI-32 (Fig. 1f). Quantification of Isl1, SMI-32, and ChAT-positive cells indicated that 70–95% of cells were positive for these MN markers, without significant differences between patients and controls (Fig. 1g). In addition, the isogenic control line was able to differentiate into MNs and expressed the MN-specific markers at a similar ratio (Supplementary Fig. 2C).

**Patient-derived MNs show FUS pathology and hypoexcitability**. We used immunostaining to visualize FUS localization in different cells types. In fibroblasts, nuclear clearance of FUS was observed in patient lines but not in healthy controls (Fig. 2a). This phenotype was prominent in fibroblasts carrying the P525L mutation, whereas it could only be detected in a few cells with the R521H mutation (Fig. 2a). In undifferentiated iPSCs, both P525L and R521H showed nuclear clearance of FUS with again a more pronounced mislocalization of FUS in the P525L mutant line (Fig. 2a). In mature MNs at day 38 of differentiation, both iPSC-derived MNs harboring the P525L or the R521H mutation showed obvious mislocalization of FUS to the neurites, whereas almost no FUS could be detected in the neurites of healthy controls (Fig. 2b). Once we corrected the point mutation, the cytoplasmic FUS localization was fully rescued (Supplementary Fig. 4A). Consistent with the observation in fibroblasts and in undifferentiated iPSCs, the P525L mutant lines showed a much higher amount of FUS mislocalization than the R521H mutant line. These results confirmed that there is indeed cytoplasmic mislocalization of the FUS protein in cells derived from FUS-ALS patients, and that the amount of cytosolic FUS depends on the nature of the point mutation. In addition, these data demonstrate that mislocalization of FUS is not MN specific as it is also observed in fibroblasts and in undifferentiated iPSCs.

We next performed an electrophysiological evaluation of iPSC-derived MNs during week 7 after initiation of differentiation. Both control and mutant MN cultures contained cells firing single (Supplementary Fig. 3A) or repetitive action potentials (Supplementary Fig. 3B) during stepwise depolarization in current clamp mode. This demonstrates that functional MN maturation from patient and control iPSC lines was not different. Although the number of electrophysiologically active cells with spontaneous action potentials revealed no differences between controls and patient MNs (Fig. 2e), a significant decrease in the frequency of spontaneous action potentials (Fig. 2f), but not in the amplitude of these spontaneous action potentials (Fig. 2g), was observed in patient-derived MNs compared with the healthy controls (Fig. 2c–g). This is in line with the hypoexcitability of ALS MNs described in previous reports[11, 13]. To investigate the $Na^+$ and $K^+$ currents in iPSC-derived MNs, we used whole-cell voltage clamp mode in a series of depolarizing steps (Supplementary Fig. 3C). After normalization for their membrane surface (pA/pF), the $Na^+$ peaks were significantly lower in patient iPSC-derived MNs compared with the controls (Supplementary Fig. 3D, F). As the $K^+$ current peaks were similar (Supplementary Fig. 3E), this suggests a lower ratio of $Na^+/K^+$ channel currents in MNs derived from mutant FUS iPSCs, which may contribute to the decreased excitability. In addition, when we recorded spontaneous post-synaptic currents in whole-cell voltage clamp mode, the frequency of synaptic activity was significantly decreased in patient iPSC-derived MNs compared with the healthy controls (Fig. 2h, j). Similar to what we observed for the spontaneous action potentials, the proportion of active cells and the amplitudes did not show any change between patients and control (Fig. 2i, k). Interestingly, the synaptic input could be blocked by the application of a glutamatergic AMPA receptor antagonist, NBQX, and not by a GABA receptor antagonist, bicuculline (Supplementary Fig. 3G, H).

**Progressive axonal transport defects in patient-derived MNs**. Disturbances in axonal transport are considered as an early event in the pathogenesis of ALS[17]. Therefore, we measured mitochondrial transport in iPSC-derived mature MN cultures in both patient and control cell lines by using MitoTracker to label mitochondria in a live cell imaging setup. Kymographs were used to quantify the total number of moving as well as stationary mitochondria. In these kymographs, stationary mitochondria are shown as vertical lines and diagonal lines represent moving ones (Fig. 3a).

Although the total number of mitochondria was not different between patient and control MNs at 6 to 7 weeks after initiation of differentiation (Fig. 3b), patient-derived MNs showed a significant decrease in the number of moving mitochondria (Fig. 3c). There was no significant difference between the different patient lines. When we measured mitochondrial transport at different times after plating, a progressive mitochondrial transport defect was observed in patient-derived MNs with a normal number of moving mitochondria at 2 weeks of differentiation and a significantly reduced number of moving mitochondria starting from the third week, which became worse over time (Fig. 3e).

**Fig. 1** Generation and characterization of iPSCs and MNs from ALS patients and controls. **a** Semi-quantitative PCR of Oct4, Nanog, Sox2, and Rex1 in patient iPSCs, control iPSCs, human ESC (ES), and human fibroblasts (HF). **b** Immunocytochemical characterization of iPSCs for the pluripotency markers Tra1-60, SSEA4, Oct-4, and Nanog from 2/2, 3/1-2, and 3/2 lines. Scale bar = 50 μm. **c** Sequencing confirming the heterozygous R521H and P525L FUS mutations in patient-derived iPSCs and the absence of FUS mutations in the control iPSC lines. **d** Hematoxylin–eosin staining identifying three germ layers in teratomas formed after iPSC injections in immunodeficient mice. Neural-like tissues represent ectoderm, cartilage represents mesoderm, and gut epithelium represents endoderm. **e** Schematic protocol of motor neuron differentiation. Y Y-27632, SB SB 431542, LDN LDN-193189, CHIR CHIR99021, RA retinoic acid, SAG smoothened agonist, DAPT a γ-secretase inhibitor, NEPs neuroepithelial stem cells, MNPs motor neuron progenitors, sMNs spinal motor neurons, BDNF brain-derived neurotrophic factor, GDNF glial cell line-derived neurotrophic factor, CNTF, ciliary neurotrophic factor. **f** Staining of different motor neuron markers (Hb9, Isl1, ChAT, SMI-32, Synapsin1), and DAPI in mutant FUS expressing and control cells at day 38. Scale bar = 50 μm. **g** Quantification of Isl1-positive (upper panel), ChAT-positive (middle panel), and SMI-32-positive (lower panel) cells expressed relative to the total number of DAPI-labeled cells. N = 10 images per line. Data are represented as mean ± SEM

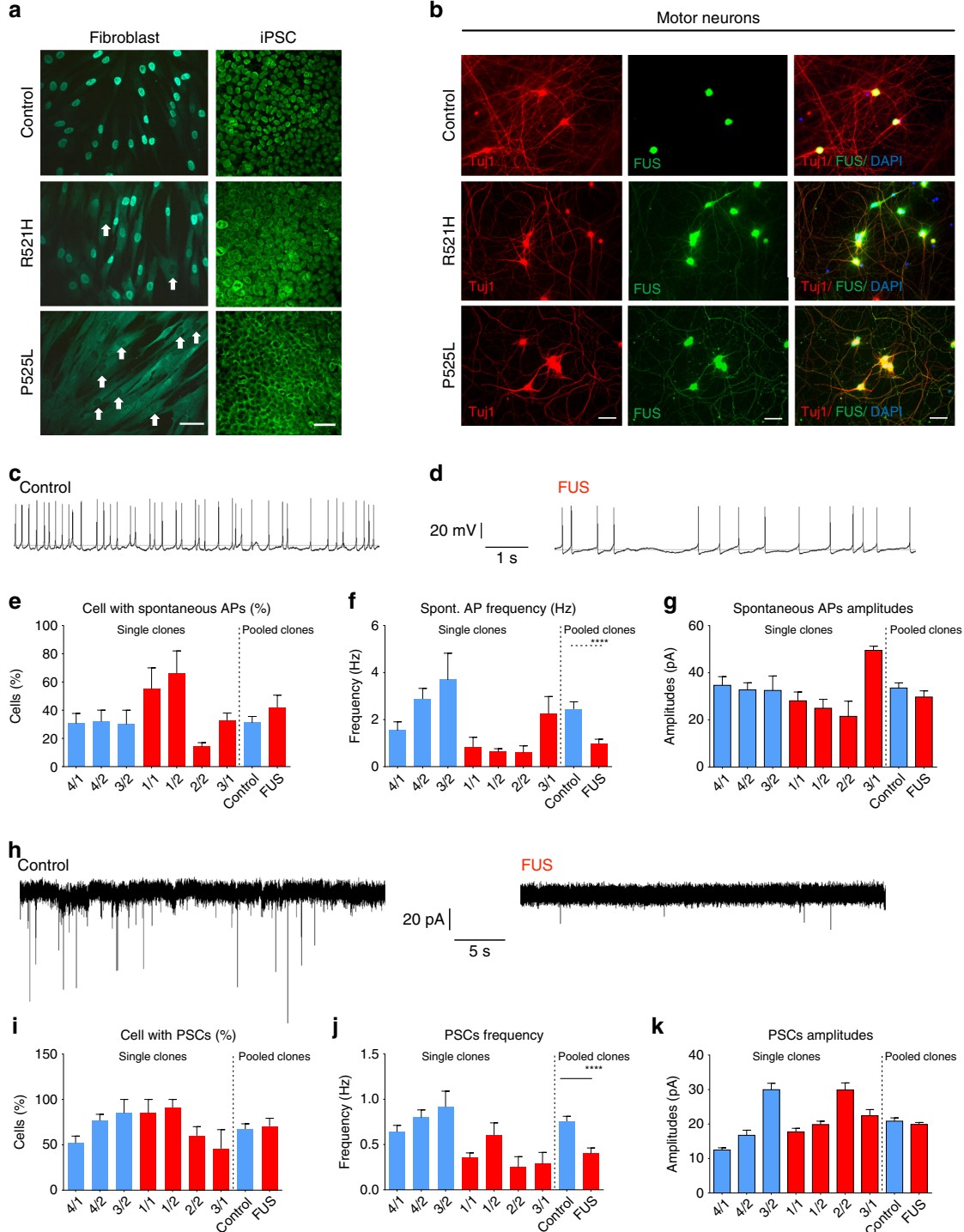

**Fig. 2** ALS patient-derived MNs exhibit cytoplasmic FUS localization and hypoexcitability. **a** Immunostaining of FUS in fibroblasts and iPSCs from ALS patients and healthy controls. Nuclear clearance of FUS was observed in patient fibroblasts. White arrows highlight the nuclear clearance in patient fibroblasts carrying the R521H or P525L mutation. Scale bar = 70 μm. **b** Immunostaining of FUS and Tuj1 with DAPI in motor neurons derived from iPSCs of ALS patients and healthy controls at the fourth week of differentiation. Scale bar = 70 μm. **c**, **d** Representative traces of spontaneously occurring action potentials (APs) in healthy control and FUS iPSC-derived MNs during the seventh week of differentiation. **e–g** Characterization of MNs differentiated from iPSCs with spontaneous APs. The relative number of cells (**e** n = 110 and n = 67 for control and patients, respectively), the frequency of spontaneous APs (**f**, n = 48 and n = 34 for control and patients, respectively, Mann–Whitney test, ****P value is 0.0001) and the amplitude (**g** n = 47 and n = 54 for control and patients, respectively) were measured. Data values represent mean ± SEM. **h** Representative traces of spontaneous postsynaptic currents (PSCs) in healthy control and FUS iPSC-derived MNs with typical MN appearance during the seventh week of differentiation. **i–k** Quantification of the relative number of cells with PSCs (**i** n = 110 and n = 67 for control and patients, respectively), the PSC frequency (**j** n = 91 and n = 57 for control and patients, respectively, Mann–Whitney test, ****P value is 0.0001) and the PSC (**k** n = 91 and n = 57 for control and patients, respectively) amplitude in iPSCs-derived motor neurons from ALS patients and healthy controls

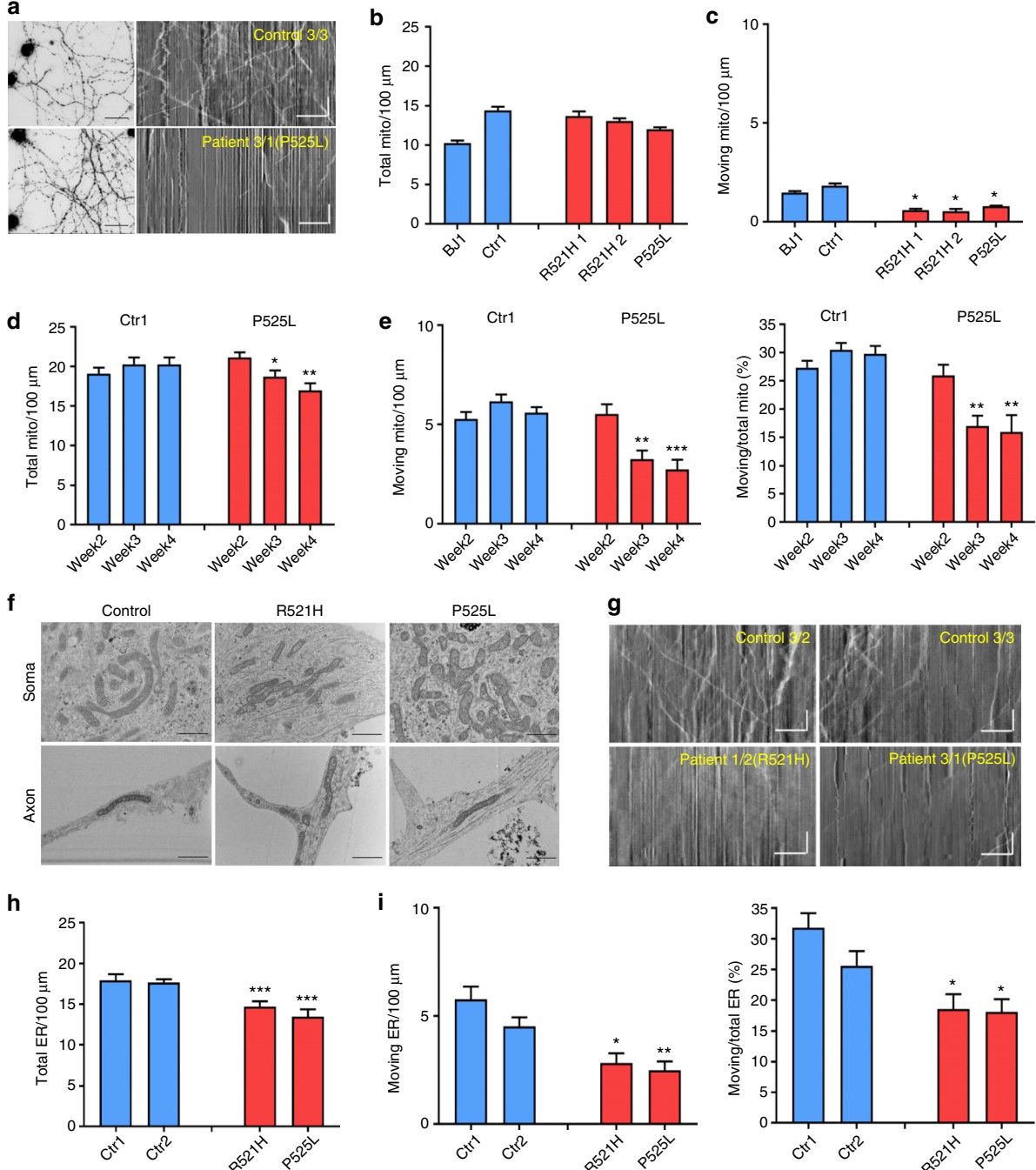

**Fig. 3** Progressive axonal transport defects in ALS patient-derived MNs. **a** Representative fluorescent micrograph from control and FUS-ALS patient-derived MNs loaded with MitoTracker-Red (left) and typical kymographs (control: 3/3; patient: 3/1) obtained from MNs (right). Stationary mitochondria are visible as straight vertical lines, while moving mitochondria are deflected as tilted lines. Scale bar time (vertical): 35 s; scale bar distance (horizontal): 25 μm. **b** Quantification of the number of stationary mitochondria normalized to a neurite length of 100 μm during 200 s for MNs derived from different iPSC lines at sixth to seventh week after starting MN differentiation (*n* = 5). Data values represent mean ± SEM. **c** Quantification of the number of moving mitochondria for MNs derived from different iPSC lines. **d** Quantification of stationary mitochondria for motor neurons derived from a patient with a P525L mutation and a control iPSC line as a function of time after plating (2 weeks, 3 weeks, 4 weeks after plating, *n* = 13 and *n* = 14 for patient and control, respectively). Data values represent mean ± SEM. **e** Quantification of moving mitochondria and ratio of moving to total mitochondria for motor neurons derived from a patient line (3/1) with a P525L mutation and a control iPSC line (3/2). **f** EM analysis of control (3/2) and patient (R521H: 2/2; P525L:3/1)-derived motor neurons at the fourth week after plating. Images of both soma and neurites are shown. Representative images from three independent experiments are presented. Scale bar = 100 nm. **g** Typical kymographs (control 1: 3/2; control 2: 3/3; R521H: 1/2; P525L: 3/1) obtained from motor neurons loaded with ER Tracker-Red. Scale bar time (vertical): 35 s; scale bar distance (horizontal): 25 μm. **h** Quantification of stationary ER vesicles from MNs derived from patients with P525L or R521H mutations and control iPSC lines at the fourth week after plating (*n* = 10, *n* = 11, *n* = 12, *n* = 11 for Ctr1, Ctr2, R521H, and P525L, respectively). Data values represent mean ± SEM. **i** Quantification of moving ER vesicles and ratio of moving to total ER vesicles for MNs derived from patients with P525L or R521H mutations and control iPSC lines (data are plotted as mean ± SEM; one-way ANOVA with post-hoc Tukey's test, *, **, ***P values of 0.05, 0.01, and 0.001, respectively). Data values represent mean ± SEM

Although the total number of mitochondria showed a slight decrease over time (Fig. 3d), the number of moving mitochondria relative to the total number remained significantly lower in MNs derived from patient iPSCs compared with the controls (Fig. 3e). As mitochondrial transport defects were previously correlated with severe mitochondrial damage in a SOD1[A4V] iPSC model[33], we used electron microscopy (EM) to assess for the integrity of the mitochondria. However, we could not detect any obvious morphological changes in the mitochondria in the soma or axons of patient-derived MNs (Fig. 3f).

We also measured axonal transport of endoplasmic reticulum (ER) vesicles using ER-tracker (Fig. 3g–i). Patient-derived MNs also showed a significant reduction of moving ER vesicles along the axons compared with MNs from healthy control iPSC lines (Fig. 3i). However, the total number of ER vesicles in axons from MNs of patient iPSC lines was significantly reduced compared with those from controls (Fig. 3h). The fact that the percentage of moving ER vesicles relative to the total number of ER vesicles was decreased in patient lines (Fig. 3i) shows that ER vesicle transport was also decreased, similar to what was observed for mitochondrial transport. Altogether, our data indicate that axonal transport in general is progressively impaired over time in FUS-ALS patient-derived MNs.

**Defects in axonal transport are caused by mutant FUS.** To confirm that the observed phenotypes were due to the mutation in FUS and to eliminate possible effects of the genetic background on these phenotypes, we established an isogenic control iPSC line for one of the patient-derived iPSC lines carrying the R521H mutation using the CRISPR-Cas9 technology. Sequencing confirmed that the point mutation was corrected (Fig. 4a). As indicated before, differentiation into MNs of this genetically corrected iPSC line was similar as for the parent line. We performed mitochondrial transport experiments to assess whether the correction of the FUS mutation also corrected the defects observed in MNs derived from the parent iPSC line. We found no reduction in the total number of mitochondria, nor in the number of moving mitochondria in MNs from the genetically corrected R521R iPSC line compared with iPSC-derived MNs from controls (Fig. 4b). Thus, the axonal transport defect appears to be directly linked to the mutation in FUS. In addition, a similar rescue also happened to the cytoplasmic FUS localization (Supplementary Fig. 4A).

To further confirm the causative relationship between mutations in FUS and the axonal transport deficits, we established three FUS overexpression H9-hESC lines by inserting a single copy of the coding region of wildtype FUS, FUS with the R521H mutation and FUS with the P525L mutation, containing an FRT-flanked Hyg/TK cassette, into the *AAVS1* locus[34, 35] via recombinase-mediated cassette exchange (Fig. 4e). We confirmed the absence or presence of the different mutations by sequencing (Fig. 4f). As the construct used contains a Tet-On system, it is possible to turn on the expression of FUS by adding doxycycline. In this way, possible effects of (mutant) FUS expression during the early stages of MN differentiation are circumvented. We added doxycycline from day 17 of MN differentiation and assessed FUS pathology and axonal transport on day 38. qRT-PCR showed an approximately threefold increase in *FUS* mRNA levels and this increase was only found after adding doxycycline (Supplementary Fig. 4B). The induction of *FUS* from the transgene inserted into the *AAVS1* locus was confirmed by qRT-PCR using primers in the 3xflag region of the *FUS* mRNA (Supplementary Fig. 4B). Overexpression of wildtype FUS and mutant FUS did not change the differentiation capability from early post mitotic MNs to mature MNs (Supplementary

Fig. 5A, B). However, a significant reduction in the number of moving mitochondria was observed after overexpression of mutant FUS, whereas wildtype FUS overexpression did not affect axonal transport (Fig. 4g). A similar situation was also observed for the cytoplasmic FUS localization (Supplementary Fig. 4C). Only if we overexpressed mutant FUS, cytoplasmic FUS was observed and the P525L mutation was always worse than the R521H mutation (Supplementary Fig. 4C). We did not observe any defect in axonal transport in control-derived MNs after knocking down FUS by using ASOs (Fig. 4c, d). All together, these overexpression and knockdown data strongly suggest that the axonal transport defects are due to mutant FUS expression through a gain-of-function mechanism.

**HDAC6 inhibitors rescue axonal transport defects.** We next performed studies to investigate the possible mechanism(s) responsible for the axonal transport defects in the iPSC-derived MNs-containing mutant FUS. As indicated before, we first assessed the mitochondrial integrity in mutant FUS and control-derived MNs by using EM. We could not identify abnormal mitochondrial morphology in the FUS mutant MNs (Fig. 3g). In addition, we did not observe any change in neurofilament light chain (NFL) expression in patient-derived MNs at the fourth week of differentiation (Supplementary Fig. 6). As both mitochondrial and ER vesicle transport showed defects in mutant FUS MNs, we wondered whether the mitochondria-associated ER membranes (MAMs) could be affected[19]. Therefore, we stained both for mitochondria and ER, and quantified the percentage of overlay of both signals. ER was labeled with protein disulfide isomerase (PDI), whereas mitochondria were labeled using translocase of the outer mitochondrial membrane protein-20 (TOM20). Pearson's correlation coefficient was used for the analysis. ALS patient-derived MNs all showed a significantly decreased overlap of ER and mitochondrial staining especially in neurites (Fig. 5a, b). MAMs also influence phospholipid metabolism. Phosphatidylcholine (PC) is the most common lipid in the cell and most enriched in the ER[19, 36]. Cells with defective MAMs have a lower rate of phosphatidylserine (PS) conversion to PC than wildtype cells[37]. Therefore, production of PC can be used to assess MAM disturbances. We harvested culture medium at different time points during the in vitro differentiation process (second, third, and fourth week after plating) and measured the level of released PC. We observed a reduction of PC in mutant FUS MNs (Supplementary Fig. 7, Fig. 5c) at the fourth week after plating. Consistent with the progressive defects found in axonal transport (Fig. 3e), PC release decreased with prolonged time in culture (Supplementary Fig. 7).

HDAC6, a class IIb histone deacetylase, is the major enzyme with α-tubulin deacetylation activity[23]. In a previous study, we discovered that HDAC6 inhibitors restore mitochondrial axonal transport defects in DRG neurons from a CMT disease mouse model, which correlated with increased α-tubulin acetylation[23]. In this study, we assessed the acetylation level of α-tubulin in MNs differentiated from a mutant FUS line and its isogenic control. Using western blot analysis, we detected a slight decrease of acetylated α-tubulin in patient-derived MNs compared with the isogenic control (Fig. 6d). Moreover, ER-sliding dynamics and MAM occur on acetylated microtubules[38]. Therefore, we investigated the effect of HDAC6 inhibition on MAMs and on ER vesicle and mitochondrial axonal transport.

First, we used Tubastatin A, which is a highly selective inhibitor of the deacetylase function of HDAC6, to treat both patient and healthy control iPSC-derived MNs[39]. The next day, cells were fixed for immunostaining or loaded with MitoTracker and ER-Tracker to measure axonal transport. On the basis of the

staining results of Tom20 and PDI, the colocalization of ER and mitochondria was significantly increased along axons in patient-derived MNs after treatment with Tubastatin A (Supplementary Fig. 9A). This indicates that MAMs were increased. In addition, the number of moving mitochondria along the axons was restored in the patient-derived MNs (Supplementary Fig. 9B). Similarly, both total number and number of moving ER vesicles were also increased to the same level as in healthy controls

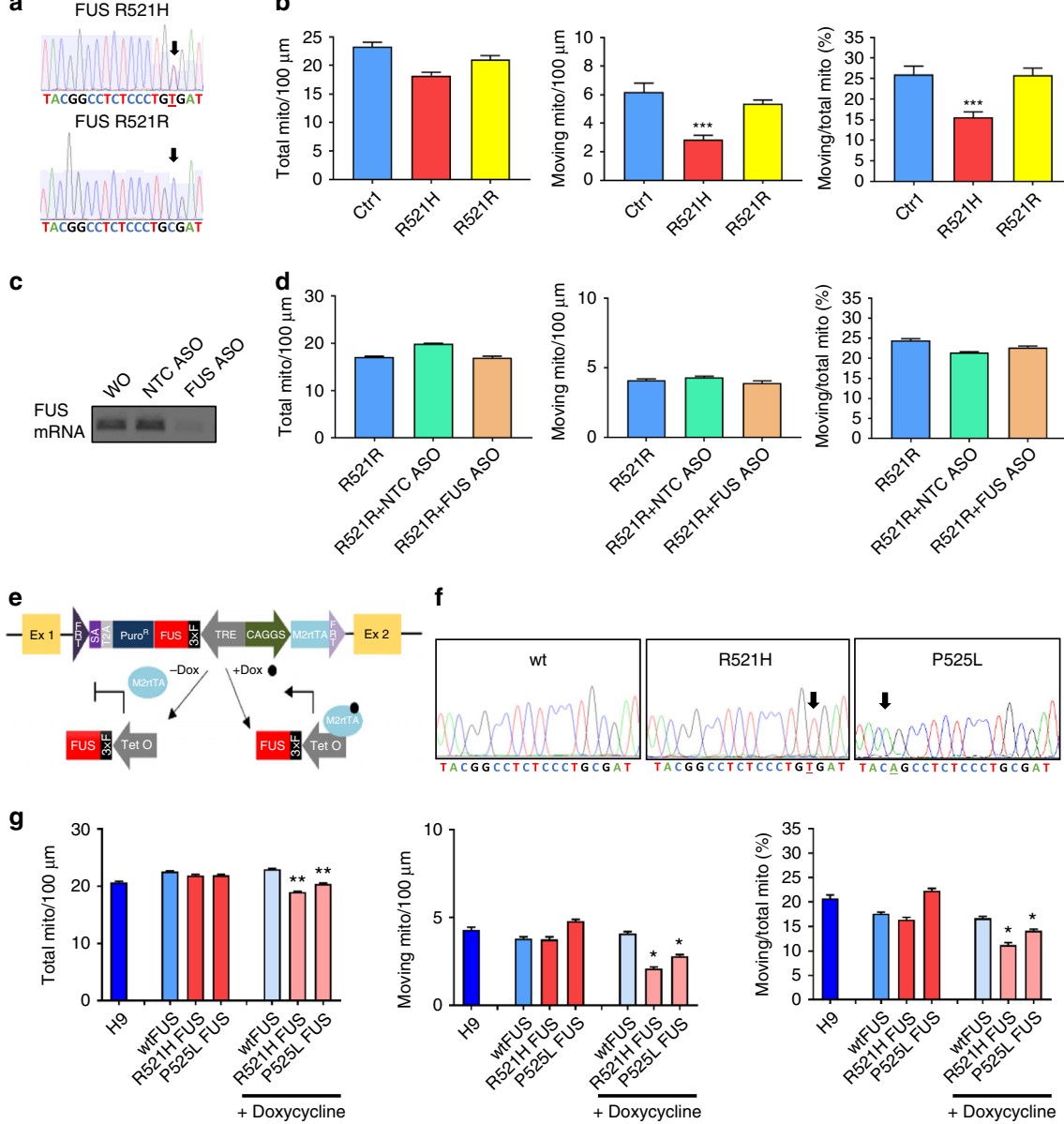

**Fig. 4** Axonal transport defects are caused by mutant FUS. **a** Genomic DNA sequencing showing the nucleotide change (C to T) in the R521H patient line (labeled as FUS R521H) and the homozygous nucleotide C in the corrected isogenic control iPSC line (labeled as FUS R521R). **b** Quantifications of total mitochondria, moving mitochondria, and ratio between moving to total mitochondria normalized to a neurite length of 100 μm during 200 s in motor neurons derived from a control iPSC line, from a patient iPSC line with the R521H mutation and its isogenic control measured at 4 weeks after plating. ($n = 11$, $n = 14$, $n = 12$, respectively, for wildtype, mutant, and isogenic control, mean ± SEM; one-way ANOVA with post-hoc Tukey's test). **c** RT-PCR validation of ASO knockdown of FUS in motor neurons. **d** Knockdown of FUS did not interfere with total number of mitochondria, moving mitochondria, and mitochondrial transport efficiency. Scrambled ASO was used as a negative control (NTC ASO). $n = 20$, $n = 19$, $n = 17$ for R521R, R521R + NTCASO, R521R + FUS ASO, respectively. Data values represent mean ± SEM. **e** Schematic diagram of the strategy used for inducible expression of FUS in the *AAVS1* site using recombinase-mediated cassette exchange (RMCE) and the targeting vector *pZ: F3-P TetOn-3 × F-FUS-F* flanked by short flippase recognition targets (FRTs) in the engineered H9-hESC line. This vector also contains a Tet-On system. Adding doxycycline (Dox) to the medium results in its binding to the M2 tetracycline transactivator (M2rTA) protein. This Dox–M2rtTA complex is capable of binding the Tet operator (TetO), which is part of the tetracycline response element (TRE), to trigger FUS expression. **f** DNA sequencing showing homozygous wildtype FUS, mutant FUS carrying the R521H or the P525L mutation in the hESC lines that can overexpress FUS. **g** Calculations of stationary mitochondria, moving mitochondria, ratio of moving to total mitochondria per 100 μm neurite length during 200 s for motor neurons derived from hESCs at the fourth week after plating. Only overexpression of mutant FUS induces mitochondrial transport defects ($n = 12$, $n = 15$, $n = 12$, $n = 13$, $n = 14$, $n = 11$, $n = 14$, respectively, for H9, wtFUS, R521H FUS, P525L FUS, wtFUS + Dox, R521H FUS + Dox, P525L FUS + Dox, mean ± SEM, one-way ANOVA with post-hoc Tukey's test, *, **, ***P values of 0.05, 0.01, and 0.001, respectively)

(Supplementary Fig. 9C). Tubastatin A had no effect on MNs derived from healthy controls (Supplementary Fig. 9).

To confirm the Tubastatin A effect, we also tested ACY-738, which is another newly developed HDAC6 inhibitor with a higher blood–brain–barrier permeability[40]. We treated the cells overnight and a similar rescue was observed in patient-derived MNs carrying different FUS mutations, whereas there was again no

effect observed in controls (Fig. 5). The overlap between ER and mitochondrial markers increased 1.5-folds in axons (Fig. 5a, b). The percentage of moving axonal mitochondria and ER vesicles in the patient cells also reversed to levels seen in the controls (Fig. 5d, e). Consistently, a slight increase in the PC level was observed in medium of patient-derived MNs after treatment with the HDAC6 inhibitors. These compounds did not change the

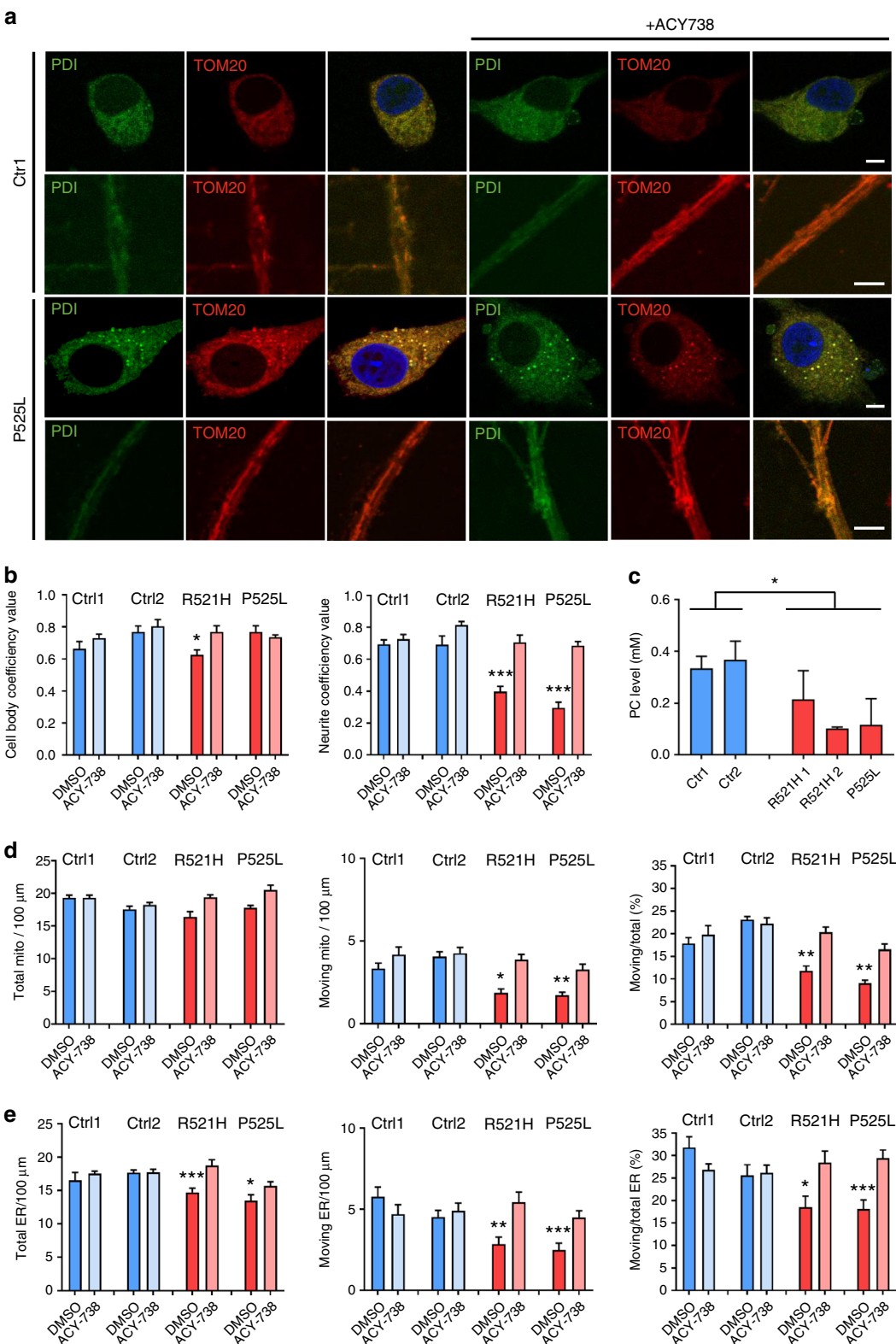

cytoplasmic FUS mislocalization in patient cells (Supplementary Fig. 8). To confirm the specificity of these HDAC6 inhibitors, we used an ASO to knockdown HDAC6 with 50% in our MNs by treating the MNs for 1 week with the ASO (Fig. 6a, b). Again, an increase of axonal transport in patient-derived MNs was observed. Similar as for the HDAC6 inhibitors, no effects were observed on MNs derived from the isogenic control (Fig. 6c).

Lastly, we investigated the effect of both Tubastatin A and ACY-738 on the acetylation of α-tubulin using western blot. An overnight treatment of our iPSC-derived MNs with both drugs markedly increased the acetylation of tubulin, both in the patient lines as well as in the isogenic control (Fig. 6d; Supplementary Fig. 10). In line with what we described before[23], there seems to be a good correlation between the increase in the acetylation level of α-tubulin and the rescue of axonal transport defects.

## Discussion

In this study, we established iPSCs from fibroblasts of four ALS patients harboring two different point mutations (R521H and P525L) in the NLS region of FUS, as well as from two healthy controls. Furthermore, we used CRISPR-Cas9 technology to correct the point mutation in one of the ALS patient-derived iPSC lines as an isogenic control line, and we overexpressed wildtype and mutant FUS in hESCs. In this way, we investigated the causative relationship between FUS mutations and the observed defects in MNs derived from these iPSCs/hESCs. In addition, we developed a highly efficient MN differentiation protocol, which generated more than 80% of cells with MN morphology, expressing typical MN markers and exhibiting typical electrophysiological characteristics. The MNs derived from these iPSCs of ALS patients exhibited typical FUS mislocalization and hypoexcitability. In addition, progressive mitochondrial and ER vesicle transport defects developed in these iPSC-derived MNs. These axonal transport defects as well as the MAM reduction could be rescued by HDAC6 inhibition and by genetic knockdown of HDAC6 using ASOs.

Consistent with previous reports using iPSC models or postmortem patient tissue[6, 7, 10–12], we observed cytoplasmic FUS mislocalization in patient cells including fibroblasts, iPSCs, and MNs. The degree of cytoplasmic mislocalization of FUS was correlated before with the mutation type and was associated with the severity of the clinical presentation and the age of disease onset[8]. This indicates that cytoplasmic FUS localization is a very early pathological change, which occurs in different cell types and of which the severity depends on the type of FUS mutation. Unlike other studies, we did not observe FUS aggregates in our mutant lines[41]. As aggregates were reported to colocalize with

stress-granule markers, we may not have detected FUS aggregates because we did not apply extra stress to the MNs[42]. It has been suggested that the mislocalization of FUS serves as a first hit in the pathophysiological cascade that makes the cell more vulnerable[6]. A second hit, such as cellular stress and/or defects in protein degradation, is proposed to assemble cytoplasmic FUS into stress granules and could trigger neurodegeneration[6].

Several studies demonstrated a number of physiological changes in different iPSC-derived ALS models. Kiskinis et al.[33] reported hyperexcitability in iPSC-derived MNs carrying SOD1, C9orf72, and FUS mutations based on increased spontaneous firing and reduced delayed rectifier K+ currents by multielectrode array recordings. Devlin et al.[15] proposed a switch from hyperexcitation to hypoexcitation of MNs, which was dependent on the differentiation time of human iPSCs obtained from patients harboring TARDBP or C9orf72 ALS mutations. Recently, Naujock et al.[13] showed hypoexcitability in MNs derived from mutant SOD1 and FUS iPSCs differentiated for 7 weeks. Although the possible change from hyperexcitability to hypoexcitability was evaluated, only increased synaptic activity but not enhanced AP firing frequencies was found in cells differentiated in culture for 3–4 weeks. The hypoexcitability was confirmed in our study as patient-derived MNs presented with a reduced frequency of APs and postsynaptic currents at week 7 of differentiation. Hypoexcitation could be a late phenotype, which can eventually cause cell death. However, we did not observe obvious cell death in the MNs derived from patient iPSCs. As a consequence, we consider the hypoexcitability as an early pathological change rather than a direct cause of cell death.

In addition, we identified axonal transport defects in MNs derived of mutant FUS iPSCs. Axonal transport abnormalities have also been reported in several animal models of ALS, especially in SOD1 models[17, 43, 44]. SOD1 mutations can affect neurofilaments, mitochondria, and vesicle transport[44]. Only one study based on an ALS iPSC-based model has previously identified mitochondrial transport defects in MNs derived from ALS patients carrying an A4V mutation in SOD1[33]. In that study, mitochondrial damage was observed based on EM and it was hypothesized to be the cause of the mitochondrial transport defects[33]. In the current study, we did not observe obvious structural changes in the mitochondria of mutant FUS MNs. In addition, ER vesicle transport along the axons was also decreased indicating that the disturbance in axonal transport was a more general problem. These axonal transport defects developed as a function of time in culture. They were significant at the third week after plating and became progressively worse when cells were maintained in culture. Defective axonal transport defects were also observed as a common phenotype in Drosophila models

**Fig. 5** Restoration of axonal transport and of ER-mitochondrial overlay by HDAC6 inhibition. **a** Immunostaining for ER (using mouse PDI antibody) and mitochondria (using rabbit TOM-20 antibody) of motor neurons derived from iPSC from patients carrying the P525L mutation and healthy controls before and after an overnight treatment with ACY-738 (1 μM). The separate views show co-localized pixels in the cell body and neurites. Scale bar = 5 μm. **b** Quantification of the proportion of ER-merged regions and mitochondrial-merged regions relative to the total mitochondrial staining. Data were analyzed by determining the Pearson's correlation coefficient using ImageJ (n = 20, mean ± SEM, one-way ANOVA with post-hoc Tukey's test, *, ***P values of 0.05 and 0.001). **c** Phosphatidylcholine (PC) level in culture media from motor neuron. Media were taken for ELISA after 2 days on the culture at the fourth week after the start of differentiation. A decrease was observed in patient-derived motor neurons. Representative experiment is shown (mean of technical duplicates ± SD). **d** Quantification of stationary mitochondria, moving mitochondria, and ratio between moving to total mitochondria normalized to a neurite length of 100 μm in motor neurons derived from patient and healthy control iPSC lines at the fourth week after plating with or without an overnight treatment with ACY-738 (n = 18, n = 16, n = 15, n = 18, n = 15, n = 13, n = 17, n = 13 for Ctr1 + DMSO, Ctr1 + ACY-738, Ctr2 + DMSO, Ctr2 + ACY-738, R521H + DMSO, R521H + ACY-738, P525L + DMSO, P525L + ACY-738, respectively, mean ± SEM, one-way ANOVA with post-hoc Tukey's test; *, **P values of 0.05 and 0.01, respectively). **e** Quantification of stationary ER vesicles, moving ER vesicles, and ratio between moving to total vesicles normalized to a neurite length of 100 μm in motor neurons derived from patient and healthy control iPSC lines at the fourth week after plating with or without an overnight treatment with ACY-738 (n = 10, n = 19, n = 11, n = 18, n = 12, n = 12, n = 11, n = 16 for Ctr1 + DMSO, Ctr1 + ACY-738, Ctr2 + DMSO, Ctr2 + ACY-738, R521H + DMSO, R521H + ACY-738, P525L + DMSO, P525L + ACY-738, respectively, mean ± SEM, one-way ANOVA with post-hoc Tukey's test; *, **, ***P values of 0.05, 0.01, and 0.001, respectively)

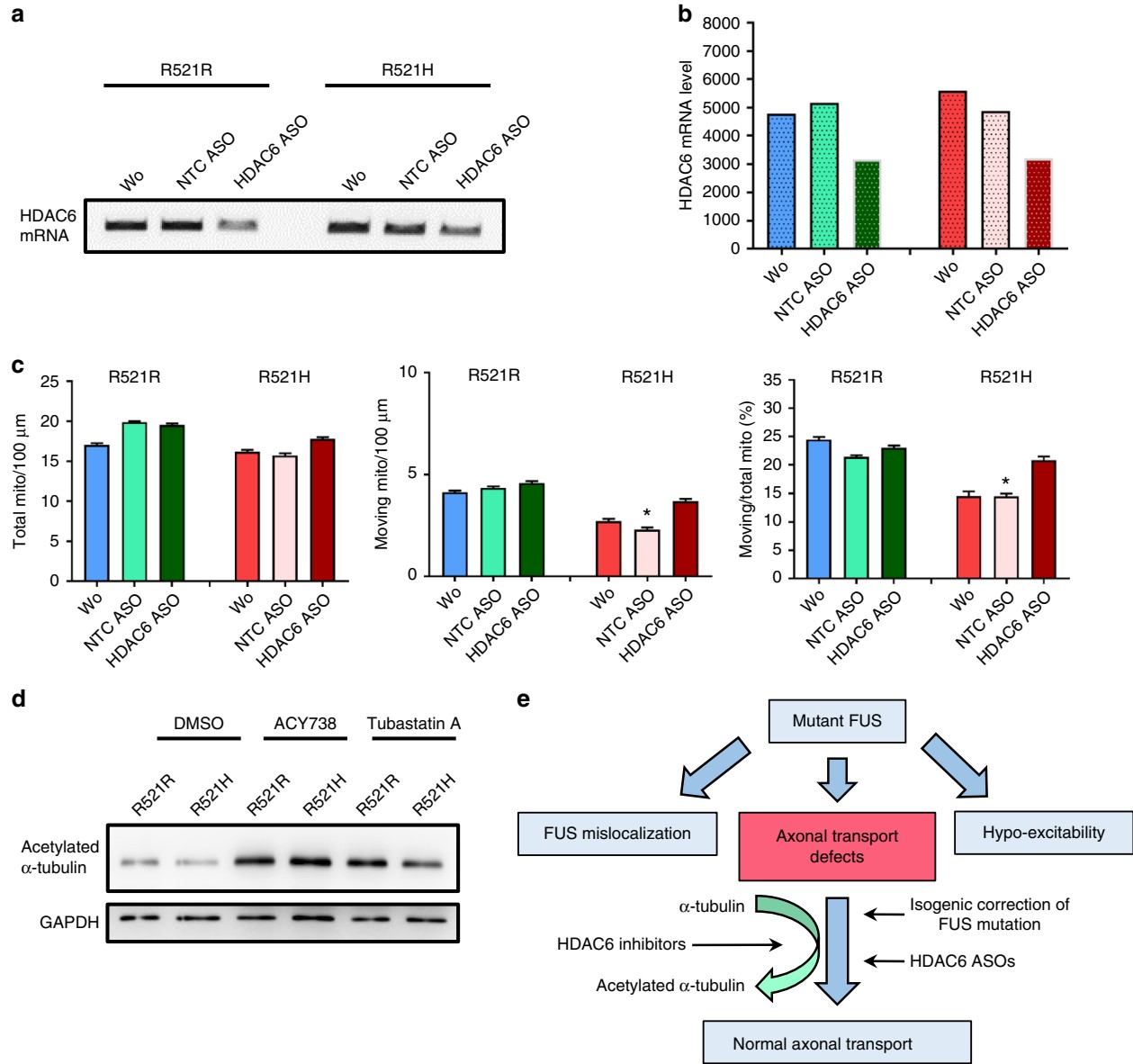

**Fig. 6** Effect of genetic HDAC6 knockdown and increased acetylation levels of α-tubulin by HDAC6 inhibition. **a** RT-PCR validation of knockdown by an antisense oligonucleotide (ASO) against HDAC6 in MNs. **b** Quantification of the RT-PCR results of the gel shown in **a** illustrating the HDAC6 knockdown by the ASO. **c** Knockdown of HDAC6 using an ASO in patient-derived MNs increased axonal transport based on tracking mitochondrial movement (total number, moving number, and ratio between total and moving mitochondria). Scrambled ASO was used as a negative control (NTC ASO). $n = 20$, $n = 19$, $n = 18$, $n = 10$, $n = 11$, $n = 10$ for R521R, R521R + NTC ASO, R521R + HDAC6 ASO, R521H, R521H + NTC ASO, R521H + HDAC6 ASO, respectively; ANOVA with post-hoc Tukey's test, *P value of 0.05. **d** Western blot from MNs (R521H mutant line 2/2 and isogenic control R521R), with and without treatment with ACY-738 or Tubastatin A at day 31 of differentiation. The blot was probed with antibodies directed to acetylated α-tubulin, and GAPDH. **e** Schematic representation of our results indicating that HDAC6 inhibition rescues axonal transport defects in FUS-iPSC-derived motor neurons through increasing acetylation levels of α-tubulin

of *TARDBP*, *FUS*, and *C9orf72*[17]. These axonal transport deficits are considered as an early pathological change during the development of ALS, which might cause dysfunction of the neuromuscular junctions, which highly rely on energy supplied by mitochondria and on efficient synaptic cargo delivery to control proper muscle contraction[45]. If axonal transport is hampered in MNs, this leads to the loss of neuromuscular junctions, axon degeneration, and finally death of the soma. This sequence of events has led to the concept of a dying back mechanism in ALS[16]. As axonal transport defects were rescued in the isogenic genetically corrected R521R iPSC-derived MNs, the mutation itself is sufficient to induce this phenotype. Moreover, the same mitochondrial transport defects were detected in MNs

generated from hESC in which two different mutant FUS genes were inducibly overexpressed from within the *AAVS1* locus, whereas wildtype FUS overexpression had no effect. This is comparable to a previous study detecting axonal branching and growth defects in primary neuronal cultures of mice overexpressing mutant FUS (P525L and R521H), whereas no defects were observed after overexpression of wildtype FUS[46]. Overexpression of caz, the *Drosophila* orthologue of human FUS in flies, did also not induce axonal transport defects, whereas caz variants mimicking mutant FUS also induced mitochondrial transport defects[17]. Both gain-of-function and loss-of-function mechanisms have been proposed as the basic mechanism underlying the pathogenic effect of mutations in FUS. Our data

strongly suggest that at least the defects in axonal transport are due to a gain of function of mutant FUS-related toxicity, which is also consistent with recent studies in a mouse model[47]. In a recent study, inhibition of the mitochondrial localization of TDP-43 can prevent mitochondrial dysfunction and neuronal loss in a TDP-43[M337V] mouse model. This indicates that there is a link between TDP-43 localization and mitochondrial movement, which may also be true for FUS[48].

To gain insights into the mechanism(s) underlying the axonal transport defects in mutant FUS MNs, we first assessed cytoskeletal changes. Although a previous study demonstrated that NFL was decreased in mutant SOD1 iPSC MNs[49], we did not observe any significant changes in the staining for NFL in both patient and control cells. Next, we investigated whether MAMs were affected by FUS mutations. MAMs have multiple functions including lipid synthesis, mitochondrial dynamics, $Ca^{2+}$ homeostasis, ER stress, intracellular trafficking, and autophagy[19]. Overexpression of TDP-43 was reported to cause MAM disruption in vivo and in vitro[21]. In a recent report, FUS was suggested to activate GSK-3β, which disrupts MAMs through a VAPB–PTPIP51 interaction[22]. By staining for both mitochondria and ER, we found that overlap of ER and mitochondrial markers was significantly lower in FUS compared with the control MNs. A limitation of our study is that we could only estimate the MAM reduction using stainings. As MAMs represent a very small region where the distance between mitochondria and ER is limited to 10–30 nm, and where only 5–20% of the mitochondrial surface is closely connected to the ER membranes[19], it is difficult to accurately quantify the percentage of MAMs. As another readout for MAM integrity, phospholipid release by FUS-mutant MNs was assessed. This demonstrated significantly decreased PC release by FUS-mutant MNs, which was reversed following genetic correction.

FUS can bind and regulate mRNAs of several motor proteins including KIF5C, KIF1B, and KIF3A[50]. All these motor proteins are involved in axonal transport of mitochondria and vesicles. KIF5C belong to the Kinesin1 family, which drives anterograde (=synapse directed) transport of mitochondria along axons[51]. An outer mitochondrial protein, Miro1, connects mitochondria with Kinesin1[51]. Miro1 preferentially localizes in MAM sites and part of the ER will be transported together with mitochondria based on this association[51]. It could be that if MAMs decrease, axonal transports will decrease because of this reduced Miro1 linkage.

We could rescue both mitochondrial and ER vesicle transport defects by using two different HDAC6 inhibitors. HDAC6 inhibition also increased the total number of ER vesicles. These compounds had no effect on healthy control-derived MNs indicating that axonal transport is already maximal and cannot be increased by inhibiting deacetylation in these cells. Knockdown of HDAC6 using ASOs also resulted in the same rescue effects on the axonal transport. This did not only confirm that HDAC6 has an important role in this rescue effect, it also confirms the specificity of the pharmacological HDAC6 inhibitors, used in our study. It was reported before that the mRNA expression of HDAC6 was regulated by a TDP-43 and FUS complex[28]. However, we did not observe any downregulation of HDAC6 expression in MNs derived of mutant FUS iPSCs. In line with the rescue of axonal transport deficits in DRG neurons from a mouse model of CMT2 and in a *Drosophila* model of Parkinson disease[23, 52], we believe that the therapeutic effect of HDAC6 inhibitors could be related to a more general effect due to the increase of the acetylation level of α-tubulin in the microtubules, rather than a (mutant) FUS-specific effect. This is also in line with the observation that genetic deletion of HDAC6 in an ALS mouse model with mutant SOD1[G93A] extended the lifespan of these mice by maintaining the integrity of the motor axon, whereas there was no effect on the disease onset[27]. Our hypothesis illustrated in Fig. 6d is that an increased acetylation of

α-tubulin increases the proportion of motor proteins binding to the microtubules, which results in more axonal transport of mitochondria and ER vesicles along the axons. This increased binding overcomes the axonal transport defects induced by different stressors, including mutations in FUS.

Both Tubastatin A and ACY-738 are considered as being specific HDAC6 inhibitors. Although Tubastatin A has a higher selectivity, the advantage of ACY-738 is that it has a better capability to cross the blood–brain–barrier and that a related compound (ACY-1215, also called ricolinostat) is in clinical trials without reported side effects until now[53]. As a consequence, pharmacological HDAC6 inhibition seems to be an interesting strategy to counteract general axonal transport defects, which correlates with an increased acetylation of α-tubulin.

In conclusion, our study showed that point mutations in *FUS* result in clear defects in MNs derived from FUS-ALS patients, which is confirmed by gene editing and overexpression approaches. Furthermore, we can rescue the defects pharmacologically by HDAC6 inhibition and by genetic silencing of HDAC6, which suggests that this could become a new therapeutic strategy for ALS.

## Methods

**Generation of iPSCs from fibroblasts**. Primary human fibroblasts were obtained from skin biopsies of ALS patients and controls with the approval of the ethical committee of the University Hospitals Leuven (S50354). The 1/1 and 2/2 samples were obtained from a 33–year-old and a 71-year-old female patient, respectively. Both of these patients carry a R521H mutation in FUS. The 3/1 sample is from a 17-year-old male patient carrying a de novo P525L FUS mutation. The 3/2 and 3/3 samples are from the unaffected parents of patient 3/1. The 2/1 sample is from a presymptomatic patient carrying a R521H mutation. Human iPSC lines were generated by using the CytoTune®-iPS 2.0 Sendai Reprogramming Kit (Invitrogen). Reprogramming factors (Kif4, Oct3/4, Sox2, and cMyc) were transfected by using non-integrating Sendai virus vectors from the kit. The clearance of these Sendai-virus vectors was confirmed by using TaqMan® iPSC Sendai Detection Kit (Invitrogen). Control BJ1 iPSC were generated from BJ fibroblasts (ATCC® CRL-2522™). The 4/1 and 4/2 are published healthy control iPSCs[13].

**Gene correction of iPSCs**. Creation of isogenic controls of iPSC carrying the R521H mutation in FUS was performed by CellSystems (Troisdorf, Germany). iPSCs were transfected with gRNA vector, Cas9 vector, and donor DNA. Transfected cells were selected with puromycin for 2 days. Single clones were genotyped with genomic DNA PCR and subsequently sequenced. The absence of the FUS mutation was confirmed by this sequencing.

**Recombinase-mediated cassette exchange in hESCs**. The FRT-containing hESC line H9 (WA09) was purchased from WiCell Research Institute (Madison, USA) and used for recombinase-mediated cassette exchange to establish the FUS hESC-overexpressing FUS. Wildtype FUS cDNA sequences were purchased from Ori-Gene Technologies (Rockville, USA). Point mutations were introduced by PCR and the cDNA was inserted into the *AgeI* and *MluI* sites of the *pZ:F3-P–Tet-On-3f-tdT-F* plasmid (constructed by the lab[34, 35]) to replace the tdT cassette to allow inducible FUS expression. The RMCE protocol used was described before[34, 35]. Puromycin (100–200 ng/ml) and 0.5 μM 1-(2-deoxy-2-fluoro-beta-D-arabinofur-anosyl)-5-iodouracil (FIAU) were used for colony selection. All PCR primers are listed in Supplementay Table 2.

**Cell cultures**. hESCs and human iPSCs were maintained on GeltrexR LDEV-Free hESC-Qualified Reduced Growth Factor Basement Membrane Matrix (A1413302, Gibco™) in Essential™ 8 medium (A1517001, Gibco™) with 1000 U/ml penicillin–streptomycin. Colonies were routinely passaged with 0.5 mM EDTA (15575-020, Invitrogen) in Dulbecco's phosphate-buffered saline (DPBS). Cultures were routinely analyzed by PCR for mycoplasma contamination.

**Teratoma formation and analysis**. iPSCs were collected through enzymatic dissociation, $5-10 \times 10^6$ cells were resuspended in 100 μl phosphate-buffered saline (PBS) and injected with 100 μl matrigel (Becton Dickinson) subcutaneously in the back of immunodeficient RAG2–/– γc–/– mice. Tumors generally developed within 4–8 weeks. Animals were killed for dissection. Teratomas were fixed overnight in 4% paraformaldehyde, washed in 70% ethanol and subsequently embedded in paraffin. After sectioning, the presence of cells from the three germ layers was assessed following hematoxylin and eosin staining. Hematoxylin–eosin staining was performed by incubating sections for 2.5 min in Harris' hematoxylin, washing with water for 10 min, and incubation for 10 s in 0.01 g/ml eosin. These experiments were approved by the Ethical Committee for animal experiments of the KU Leuven.

**Differentiation of MNs from iPSCs or hESC**. Motor neuron differentiation was performed as described before, with some modifications[32]. iPSC clones were treated with collagenase type IV to form small clusters and resuspended in Essential™ 8 medium. During the first 2 days, medium was changed every day with Neuronal basic medium (DMEM/F12 plus Neurobasal medium with N2 and B27 supplement without vitamin A) supplemented with 40 μM SB431542 (Tocris Bioscience), 0.2 μM LDN-193189 (Stemgent), 3 μM CHIR99021 (Tocris Bioscience), and 5 μM Y-27632 (Merck Millipore). From day 3 on, 0.1 μM retinoic acid (Sigma) and 500 nM SAG (Merck Millipore) was added. From day 8 on, BDNF (10 ng/ml, Peprotech) and GDNF (10 ng/ml, Peprotech) were added. DAPT (20 μM, Tocris Bioscience) was added on day 9. Floating clusters were dissociated into single cells for plating on day 11 by using 0.05% trypsin (Gibco™). Motor neuron progenitors were subsequently plated on laminin (20 μg/ml)-coated 12-well plates at $0.5–2 \times 10^5$ cells per well. From day 17 on, the cells were switched to motor neuron maturation medium supplemented with BDNF, GDNF, and CNTF (each 10 ng/ml, Peprotech) to keep long term cultures. Media were changed every other day by replacing half of the medium. For rescue experiments, motor neurons were treated overnight with either 1 μM Tubastatin A (Sigma), 1 μM ACY-738 (Acetylon Pharmaceuticals Inc., Boston, USA) or an equivalent amount of DMSO.

**Unassisted delivery of ASOs**. Locked Nucleic Acid (LNA™) oligonucleotides for FUS and HDAC6 were purchased from Exiqon (Vedbaek, Denmark), Motor neuron progenitors were seeded at low plating density of $0.5 \times 10^5$ cells per well (12 well plate) at day 10 of motor neuron differentiation. At day 25, oligonucleotides dissolved in sterilized water were added and mixed. The LNA oligonucleotides were used at a final concentration of 50 nM. The total incubation time before cell lysis or axonal transport analysis was 1 week.

**Patch-clamp experiments**. Patch-clamp recordings of iPSC-derived MNs were performed at room temperature with an inverted microscope (Zeiss) and an EPC-10 amplifier (HEKA)[13]. Whole-cell recordings were low-pass filtered and digitized at 2.9 and 10 kHz, respectively. Patch Master software (HEKA) was used for recording and for the final evaluation of the data. Borosilicate glass pipettes (Science Products) were pulled and polished to yield a resistance of 3–4 MOhm when filled with the internal solution (153 mM KCl, 1 mM MgCl₂, 10 mM HEPES, 5 mM EGTA, and 2 mM Mg-ATP, calibrated to pH 7.3 with KOH; 305 mOsm). Pharmacological compounds were applied to the external bath solution (142 mM NaCl, 8 mM KCl, 1 mM CaCl₂, 6 mM MgCl₂, 10 mM glucose, and 10 mM HEPES, calibrated to pH 7.4 with NaOH; 325 mOsm). NBQX (10 μM, Sigma) or bicuculline (10 μM, Sigma) were applied via gravity through the modified SF-77B perfusion fast-step system (Warner Instruments).

**Real-time PCR and sequence analysis**. RNA was isolated using an RNeasy kit (Qiagen) and reverse transcription was performed using SuperScript® III First-Strand Synthesis SuperMix for qRT-PCR (Invitrogen). Quantitative RT-PCR was performed using SYBR® Green PCR Master Mix (Applied Biosystems™) on the 7500 Step OnePlus™ Real-Time PCR System (Applied Biosystems™). All samples were run in triplicate and relative quantification was done using the ΔΔCt method with normalization to GAPDH. A list of primers can be found in Supplementary Table 2. Genomic DNA was isolated using a DNeasy kit (Qiagen) and sequencing was performed by LGC Genomics (Teddington, UK). The TaqMan® hPSC Scorecard™ Panel (Life Technologies—A15870—HPSC scorecard panel 384) kit was used to determine the expression of markers from the three germ layers.

**Western blotting and ELISA**. Cells were manually collected on ice and fresh-frozen cell samples were maintained at −80 °C until further processing. Samples were hydrolyzed in RIPA buffer (containing 50 mM Tris, 150 mM NaCl, 1% (vol/vol) NP40, 0.5% sodium deoxycholate (wt/vol), 0.1% SDS (wt/vol) complemented with protease inhibitors (Complete, Roche Diagnostics, pH 7.6). Protein concentrations were determined using the microBCA kit (Thermo Fisher Scientific Inc.) according to the manufacturer's instructions. Western blotting was performed as described before[23]. Optical densities were determined using the integrated density measurement tool of ImageJ (NIH).

For ELISA, the Phosphatidylcholine Colorimetric/Fluorometric Assay Kit (BioVision) was used to detect phosphatidylcholine level in motor neuron culture medium according to the manufacturer's instructions. The medium was collected after 2 days incubation with cells. No dilution was needed for the measurements. Absorbance was measured at 540 nm. Because of the large variation in absolute values between different batches, representative experiments are shown.

**Immunocytochemistry**. Cells plated on coverslips were fixed in 4% paraformaldehyde for 20 min at room temperature and were washed with PBS. Permeabilization and blocking was done for 30 min using PBS containing 0.2% Triton X-100 (Acros Organics) and 5% donkey serum (Sigma) for 1 h. Cells were incubated overnight at 4 °C in blocking buffer (2% donkey serum) containing the different primary antibodies (Abs). After washing with PBS, cells were incubated

with secondary Abs (Invitrogen) for 1 h at room temperature. Primary Abs and dilutions are listed in Supplementary Table 3.

Fluorescent and bright field micrographs were captured using a Zeiss Axio Imager M1 microscope (Carl Zeiss) equipped with an AxioCam MRc5 (bright field, Carl Zeiss) or a monochrome AxioCam Mrm camera (fluorescence, Carl Zeiss). Images were analyzed using ImageJ with Coloc2 plug-in. Pearson's correlation coefficients were used for expressing the intensity correlation for colocalization.

**Axonal transport analysis**. Motor neurons (day 24, 31, and 38) from patients and controls were loaded with MitoTracker-Red (50 nM, Invitrogen) or ER Tracker-Red (1 μM, Invitrogen), washed and left to equilibrate (20 min) in motor neuron maturation medium, before transferring them to a HEPES buffered salt solution (pH 7.4, 150 mM NaCl, 5 mM KCl, 1 mM MgCl₂, 2 mM CaCl₂, 10 mM glucose, 10 mM HEPES). Measurements were performed on an inverted Zeiss Axiovert 200 M microscope (Carl Zeiss) with a ×40 water immersion lens.

Motor neurons were selected under differential interference optics (DIC) based on typical morphology consisting of a soma and long-extended neurites. Both MitoTracker-RED and ER Tracker-Red were excited at 580 nm, using a TILL Poly V light source (TILL Photonics) and image sequences were recorded (200 images at 1 Hz) onto a cooled CCD camera (PCO Sensicam-QE) using TillVisION (TILL Photonics) software. A heated gravity-fed perfusion system was used to keep cells at 36 ± 0.5 °C during the recordings.

All image analysis was performed in Igor Pro (Wavemetrics) using custom-written routines based on a previously described analysis algorithm[54]. In brief, kymographs or spatio-temporal maps were constructed for each of the neuronal processes. In these maps, stationary mitochondria appear as vertical lines and moving mitochondria generate tilted lines. Proportions of moving and stationary mitochondria were extracted from the maps by marking and analyzing the properties of each of the mitochondrial trajectories.

**Electron microscopy**. MNs were plated on plastic coverslips on the tenth day of differentiation. On day 38, cells were fixed and processed for embedding and ultrathin sectioning. Subsequently, the samples were contrasted with uranyl acetate and lead citrate as described before[55]. MNs were identified by their morphology and were analyzed by using a JEOL JEM1400 transmission electron microscope (JEOL, Tokyo, Japan). Digital acquisitions were taken by a numeric camera (Quemesa; Soft Imaging System, Berlin, Germany).

**Statistical analysis**. A minimum of three independent experiments based on three different differentiation batches was always performed. Statistical analysis was performed using Graphpad Prism version 5.0b.

Mann–Whitney test were used as statistical analysis for patch clamp data. One-way ANOVA was used for the other experiments with post-hoc Tukey's test to determine statistical differences between groups.*$P < 0.05$, **$P < 0.01$, ***$P < 0.001$, ****$P < 0.0001$ were considered significant. Data values represent mean ± SEM, unless indicated otherwise.

**Data availability**. The data that support the findings of this study are available from the corresponding authors upon reasonable request.

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

## Acknowledgements

We thank Sarah Debray and Joni Vanneste for their help with iPSC culturing. We thank Balazs Toth for the initial guidance with the patch clamp experiments. We thank Franziska Bursch for helping with the patch clamp experiments. We thank Steven Boeynaems, Elke Bogaert and Benjamin Gille for guidance with the data analysis. We thank Dr Haibo Wang and Prof Muralidhar L. Hegde for warm help and nice discussions. This work was supported by the KU Leuven (C1 and "Opening the Future" Fund), the "Fund for Scientific Research Flanders" (FWO-Vlaanderen), the Agency for Innovation by Science and Technology (IWT; SBO-iPSCAF), the Belgian Government (Interuniversity Attraction Poles Programme P7/16 initiated by the Belgian Federal Science Policy Office), the Thierry Latran Foundation, the "Association Belge contre les Maladies neuro-Musculaires" (ABMM), the FWO-Vlaanderen under the frame of E-RARE-2, the ERA-Net for Research on Rare Diseases (PYRAMID), the EU Joint Programme - Neurodegenerative Disease Research (JPND) projects (STRENGTH and RiMod-FTD), the ALS Liga België (A Cure for ALS) and the Flemish government-initiated Flanders Impulse Program on Networks for Dementia Research (VIND). W.G. is supported by the China Scholarship Council (CSC). W.R. is supported through the E.

von Behring Chair for Neuromuscular and Neurodegenerative Disorders, Geneeskundige Stichting Koningin Elisabeth (G.S.K.E.) and the European Research Council under the European's Seventh Framework Programme (FP7/2007-2013)/ ERC grant agreement No. 340429. W.B. is a postdoctoral fellow of FWO. P.V.D holds a senior clinical investigatorship of FWO-Vlaanderen. F.W. and S.P. are supported in part by the Deutsche Gesellschaft für Muskelkranke and the Initiative Therapieforschung ALS e.V. including the contribution of Elmar Siger in memory of his wife Ingrid. D.B. thanks ANR (LabEx Revive, Investissement d'Avenir, ANR-10-LABX-73) and the Association Française contre les Myopathies (AFM grant 16465) for their financial support. C.L. is a PhD fellow supported by ANR-10-LABX-73.

## Author contributions

W.G. planned and performed most of the experiments. M.N. planned and performed the patch clamp measurements. L.F. helped with the iPSC differentiations and organized the schematic graph of MN differentiation. P.B. provided support for the electron microscopy experiments. L.O. and R.B. helped with the construction of inducible hESC lines. A.P., M.W. and Th.V. helped with the iPSC technology. V.B., W.B. and P.V.B. helped with the axonal transport measurements and P.V.B. wrote the axonal transport analysis software. M.J. provided ACY-738. T.T. and N.G. provided technical support, Jo.S. helped with the images and Ti.V. was involved in the ASO experiments. P.V.D. provided fibroblasts and ideas for the project, F.W. and S.P. supervised part of the iPSC differentiation and the patch clamp recordings of motor neurons, Ja. S contributed to the additional iPSC control lines in the patch clamp recordings. F.W., S.P. and W.R. provided ideas for the project. D.B. and C.L. provided the motor neuron differentiation protocol. L.V.D.B. and C.V. planned and supervised the project. W.G. and L.V. D.B. wrote the paper.

## Additional information

**Competing interests:** The authors declare no competing financial interests.

