## [Peer Review File · Nature Communications]

Editorial Note: Parts of this peer review file have been redacted as indicated to remove third-party material where no permission to publish could be obtained.

Reviewers' comments:

Reviewer #1 (Remarks to the Author):

This is a well-written manuscript with two important key messages. First, mutations in FUS result in axonal transport defects in motor neurons, specifically the transport of mitochondria. Second, HDAC6 inhibition reverses this defect. Either one of these alone would be sufficient to make this article of great interest to neuroscientists, but the authors have provided a tour-de-force by linking the two together. As they rightly say, this opens a new and important therapeutic avenue for ALS, as well as perhaps explaining FUS pathogenicity. The experiments are presented in a forthright and logical manner with considerable attention to detail. Notably they have been assiduous in quality control that gives confidence to their results. The use of multiple mutations, multiple different human sources for one of the mutations, and the use of the overexpression cassettes are strengths to the study.

I have only a few comments to make, all and they are only incremental to the data that has already been presented. Thus, they should be considered as minor suggestions.

- Consider providing a nice graph/drawing showing your proposed model
- A key concern with using HDAC6 inhibitors in humans is that this might be a widespread cellular process and thus the drugs may be associated with significant side-effects. What is known about the toxicity of the HDAC6 inhibitor that is being used as a chemotherapeutic agent?
- Similarly, how much of the HDAC6 inhibitors had to be added to cell culture to achieve these beneficial effects? What I'm trying to get at here was whether a relatively large amount had to be added to reverse the FUS effect, or was it shown at a lower concentration?
- Did you consider genetic ablation of HDAC6 as opposed to pharmacological inhibition? There is good rationale to try the pharmaceutical agents as it is these that might be used in clinical trials. Nevertheless, one does worry about off target effects of these drugs. Perhaps this concern can be assuaged by providing more information on the specificity of the drugs (rather than just saying they are very specific).
- The supplemental table/data is a bit confusing. A legend, title or key should be added to place this data in context.

Reviewer #2 (Remarks to the Author):

In this study, the authors investigate the pathological mechanisms associated with FUS-related ALS and test potential therapeutic strategies. They generated iPSCs from ALS patients carrying different FUS mutations, healthy controls and differentiated them into motor neurons. For some experiments, they also used iPSC lines in which the mutation was corrected. They found that neurons expressing mutant FUS show cytoplasmic accumulation of FUS, hypo-excitability, synaptic and axonal transport defects, consistent with many previous reports. They further demonstrated that axonal transport defects could be rescued by HDAC6 inhibitors. This manuscript is well written and easy to follow. However, the quality of the figures needs to be improved. For most panels, it is very difficult to read the labels. The main concern about this work is that, although potentially interesting, it does not provide sufficient mechanistic insight. The link between FUS and HDAC6 has already been reported before, and the authors did not investigate in detail the mechanism by which HDAC6 inhibitors suppressed axonal transport defects cause by mutant FUS.

Specific points:

1. Figure 2 A and B: authors need to improve the quality of the images. The images of iPSCs and neurons are not of sufficient clarity to support the author's conclusions. It looks like FUS is also in the cell body of control neurons?!
2. Figure 2 E – K: only one control line was used so it is difficult to draw meaningful conclusions. Other control lines should be used.
3. The total number of mitochondria at 6-7 weeks is half of the number at 2-4 weeks (Figure 3 B

and D) so there should be a decrease in mitochondria over time even in control neurons. The authors should extend their time-course to 6-7 weeks after plating to clarify this point. The same applies to the number of moving mitochondria (Figure 3 C and D). Why are the mitochondria moving less at the later time point (in both control and patient neurons)?

4. Quantification of the number of neurons positive for different markers should be performed to show that a more than 3-fold increase in FUS expression does not affect neuronal differentiation capacity.

5. Images in Figure 5 A-B are not of sufficient resolution/quality to determine co-localization. PDI and TOM20 should not be present in the nucleus of the neuron. It's not clear how this quantification was performed.

6. How do the authors explain the effect of the HDAC6 inhibitors, as no defects in acetylation were found in the patient neurons? Is the effect specific to HDAC6 activity? Additional experiments, such as CRISPR and RNAi knock-down of HDAC6 are needed to clarify this point. How much inhibition is needed to observe a protective effect?

7. Does inhibition/KD of HDAC6 affect the phosphatidylcholine levels in motor neurons?

8. What are the consequences of the axonal transport defects observed? Do they cause cell death? Can it be modulated by HDAC6 inhibition/KD?

Minor points:

- Table S1: what is the age at biopsy of patient 2/1?

- Figure S2B: which genes were analyzed by qPCR?

- on page 11: "Fig. 5C and D" should be "5D and E".

Reviewer #3 (Remarks to the Author):

Guo et al. have investigated the relationship between FUS mutations and axonal transport defects in iPS-derived motor neurons carrying ALS-related FUS mutations. They also observe that HDAC6 inhibition restores a normal axonal transport in mutant FUS iPS-derived motor neurons and propose that HDAC6 inhibitors may represent a new therapeutic strategy in ALS linked to FUS mutations.

The authors have generated iPS cells from ALS patients (R521H, P525L) and control fibroblasts and used CRISPR/Cas9 method to correct one line (R521H) into isogenic line carrying R521R. They determined that R521H and P525L- FUS proteins are mislocalized in the cytoplasm of fibroblasts, undifferentiated and four weeks differentiated iPS-derived motor neurons. They also show that mutant FUS motor neurons exhibit hypo-excitability, which is consistent with a previous report (Naujock et al., 2016). They propose that mutant FUS motor neurons present progressive axonal transport defects by studying the mobility of mitochondria and endoplasmic reticulum. These results are consistent with a previous report of axonal transport defect in *Drosophila* carrying FUS mutations (Baldwin et al., 2016). By using corrected isogenic line R521R, the authors established the causative link between FUS mutations and the axonal transport defect. In addition, they show a decrease of mitochondria-associated ER membranes (MAM) in neurites of mutant FUS motor neurons. MAMs are also involved in the regulation of the phospholipid metabolism and reduced phosphatidylcholine (PC) levels were found in culture medium from mutant motor neurons.

Finally, the authors have tested the potential of HDAC6 in reversing abnormal phenotype in FUS mutant iPS-derived motor neurons. Indeed, They had previously reported that abnormal axonal transport in a mouse model expressing mutant HSPB1-induced Charcot-Marie-Tooth disease (CMT2) could be restored by inhibition of HDAC6 (d'Ydewalle, C. et al., 2011) through an increase of α -tubulin acetylating activity. Two years later, Taes et al., have also shown that Hdac6 deletion delays disease progression in the SOD1G93A mouse model of ALS (Taes et al., 2013). While no defect of α -tubulin acetylation was detected in mutant FUS motor neurons, the authors determined that treatment by HDAC6 inhibitors such as tubastatin A and ACY738 restored both MAM and axonal transport defects. ACY738 is already in clinical trials for cancer and has a blood-brain-barrier permeability (Falkenberg et al., 2014; Majid et al., 2015), opening the opportunity for

development of a new therapeutic strategy in ALS.

The link between FUS mutations and axonal transport was already established (Baldwin et al., 2016), and the novelty comes mostly from the system used: iPS-derived motor neurons and isogenic corrected cell lines. While HDAC6 inhibitor was already tested in CMT2 and the deletion of Hdac6 was promising in ALS-SOD1 mouse model, HDAC6 inhibitors were never tested in ALS-FUS model. Hence the most important finding of this study is the effect of HDAC6 inhibitors in rescuing transport defect associated to ALS-FUS mutations in less than 24 hours.

Overall, this study uses innovative tools (iPS-derived motor neurons; CRISPR edited control line and inducible ES lines overexpressing mutant and wild-type) and confirms a role for transport deficit in FUS-ALS disease. Importantly, the authors identify a potential new therapeutic target in FUS-ALS. In this reviewer's view the tools and findings from this study are of high relevance for the field, however the quality of the evidence is often difficult to appreciate and several concerns should be addressed to reach the level required for publication in Nature Communication.

Majors concerns:

1. A general concern is the presentation of the data. Indeed, Figures and fonts are extremely small and often not readable on a printed version. Most importantly images provided on Figures 1F, 2A-B, 3A, 3F, 5A, and Supplementary Figures S2C and S8A are not appropriate (low magnification; deem staining) and results difficult to evaluate.
2. The authors show that axonal transport defects are rescued by only 24h of treatment with HDAC6 inhibitors. It would be important to demonstrate that HDAC6 inhibitors also rescue other phenotypes identified in FUS motor neurons. Indeed, the authors should perform electrophysiological experiment on FUS motor neurons treated with Tubastatin A or ACY-738 in order to assess whether HDAC6 inhibitors can rescue the hypo-excitability of FUS motor neurons. They should also test whether Tubastatin A or ACY-738 treatment restore the level of phosphatidylcholine in FUS motor neurons.
3. The authors should discuss whether HDAC6 inhibition is associated with toxicity and whether treatment longer than 24h is tolerated by FUS mutant motor neurons in culture.
4. It is also not clear whether the authors have identified anterograde or retrograde transport deficits. Most importantly, while they propose that FUS motor neurons have an axonal transport deficit their analysis does not seem to selectively monitor axons.
5. In Figure 3, the transport of mitochondria was measured in control and FUS mutant motor neurons. There is inconsistency between Figure 3E - with 5 moving mitochondria per 100um neurite length after 2, 3 or 4 weeks of differentiation - and Figure 3C with 2 moving mitochondria per 100um neurite length at 6-7 weeks of differentiation. Is there a progressive reduction of mitochondrial transport in control motor neurons between 4 to 6 weeks of differentiation?
6. In Figure 1F and S2C it is almost impossible to see the Dapi staining and to appreciate the proportion of positive motor neurons.
7. Figure 2A shows increased cytoplasmic staining in mutant fibroblasts and motor neurons but not "nuclear clearance" as claimed on page 6.
8. It is not clear why only one control line was included in the electrophysiology studies on Figure 2. Considering the important variability of the results between lines it seems important not to rely only on 1 control.
9. The result on Fig 2K should be clarified. It seems that there is a reduction of PSCs amplitudes

except for 1 mutant line. Also the error bar on the patient "pooled clones" is surprisingly small considering the variability between individual clones.

10. The quality/size of Figure 3F does not allow the readers to appreciate mitochondrial integrity. Similarly, the identification of mitochondria-associated ER membranes (MAMs) by co-staining of Tom20 and PDI can not be appreciated on Fig 5A (the colocalization is not clear even in the small zoom views).

11. The mutant FUS motor neurons do not present an increase of alpha-tubulin acetylation, while HDAC6 inhibitors restore mitochondrial and ER transport defect. The authors should better discuss the mechanism of action of HDAC6 inhibitors on axonal transport defect. The rationale to test HDAC6 inhibitors should also be included in the introduction.

12. The authors should discuss previous reports that TDP43 and FUS bind and regulate the level of HDAC6 mRNA (Kim JBC 2010; Fiesel EMBO 2010; Fiesel Mol Neurod 2011; Tollervey Nat neurosc 2011; Polymenidou Nat Neurosc 2011; Miskiewicz cell Rep 2014; Odagiri BBRs 2013).

13. The authors should better discuss the results indicating that there might be an inter-regulation between AMPA receptor and FUS (Supplemental Fig 3G, H).

14. The authors propose that the phenotypes observed are due to a gain of toxic function. However, to make this claim they would need to knock-down FUS in iPSC-derived motor neurons and measure transport and excitability properties.

Minor concerns:

1. It would be interesting to show the cellular localization of FUS in the isogenic cell line R21R and in hESC lines overexpressing mutant and wild-type FUS.

2. The following references should be included in the manuscript:

- Taes et al. HMG (2013). Hdac6 deletion delays disease progression in the SOD1G93A mouse model of ALS.

- Wang et al. Mol Ther (2017). Motor-Coordination and Cognitive Dysfunction Caused by Mutant TDP-43 Could Be Reversed by Inhibiting Its Mitochondrial Localization.

- Vance et al. Science (2009) should be included along with Kwiatkowski et al. Science (2009).

- Appropriate references should be included to the following sentence on page 3: "in iPSC models, cytoplasmic mislocalization of mutant FUS was reported by three independent groups in their patient-derived motor neurons."

3. On page 23, Reference 12 lacks the year of publication : Devlin et al. Human iPSC-derived motoneurons harbouring TARDBP or C9ORF72 ALS mutations are dysfunctional despite maintaining viability. Nat. Commun. 6, 1–12 (1AD).

4. On page 5, the sentence "the origin of the iPSCs was confirmed by SNP analysis (Suppl Table 2)" should be removed since Suppl Table 2 does not correspond to SNP analysis. It is not clear if the authors want to include or not this analysis (provided at the end without a table number and referred as "data not shown" in the text).

5. On page 8, "Fig 3G" should be removed from the sentence "change in mitochondria in the soma or axons of patient-derived MNs compared to controls (Fig. 3F, 3G)", and added to the sentence "We also measured axonal transport of ER vesicles using ER-Tracker (Fig. 3H, 3I)."

Answers to the reviewers:

Reviewer #1:

This is a well-written manuscript with two important key messages. First, mutations in FUS result in axonal transport defects in motor neurons, specifically the transport of mitochondria. Second, HDAC6 inhibition reverses this defect. Either one of these alone would be sufficient to make this article of great interest to neuroscientists, but the authors have provided a tour-de-force by linking the two together. As they rightly say, this opens a new and important therapeutic avenue for ALS, as well as perhaps explaining FUS pathogenicity. The experiments are presented in a forthright and logical manner with considerable attention to detail. Notably they have been assiduous in quality control that gives confidence to their results. The use of multiple mutations, multiple different human sources for one of the mutations, and the use of the overexpression cassettes are strengths to the study. I have only a few comments to make, all and they are only incremental to the data that has already been presented. Thus, they should be considered as minor suggestions.

Response: We acknowledge this appreciation and overall positive review.

- Consider providing a nice graph/drawing showing your proposed model

Response: This is a great suggestion and we have incorporated a scheme in Figure 6.

- A key concern with using HDAC6 inhibitors in humans is that this might be a widespread cellular process and thus the drugs may be associated with significant side-effects. What is known about the toxicity of the HDAC6 inhibitor that is being used as a chemotherapeutic agent?

Response:

[Redacted]

- Similarly, how much of the HDAC6 inhibitors had to be added to cell culture to achieve these beneficial effects? What I'm trying to get at here was whether a relatively large amount had to be added to reverse the FUS effect, or was it shown at a lower concentration?

Response: As indicated in the legend of Figure 5 and Sup. Figure 8, we always used 1 μ M for both tubastatin A and ACY-738. We know that the drugs are not yet toxic *in vitro* at a concentration of 10 μ M (results not shown). We never performed a dose-response experiment for the effect on axonal transport. The major reasons being that a very large number of differentiated cultures are needed and that the analysis of the axonal transport data is extremely time consuming. However, we have already published dose-response curves for the effect of both tubastatin and ACY-738¹ on the acetylation level of α -tubulin in cultured cells. As there seems to be a good correlation between the effect on α -tubulin acetylation and the positive effect on axonal transport, our previous data suggest that rescue effects can also be obtained at lower concentrations (we already observed effects on acetylation with concentrations as low as 100 nM). We always selected 1 μ M as this resulted in the maximal effect on α -tubulin acetylation.

- Did you consider genetic ablation of HDAC6 as opposed to pharmacological inhibition? There is good rationale to try the pharmaceutical agents as it is these that might be used in clinical trials. Nevertheless, one does worry about off target effects of these drugs. Perhaps this concern can be assuaged by providing more information on the specificity of the drugs (rather than just saying they are very specific).

Response: We fully agree with this constructive suggestion. During the revision, we tried different strategies to influence the expression level of HDAC6. We used lentiviruses

expressing shRNAs, the CRISPR/Cas9 technique and antisense oligonucleotides (ASOs) to knock down HDAC6 in our cultured motor neurons. The lentiviral-shRNA and CRISPR/Cas9 did not give us the expected knock down and/or resulted in toxicity. However, treatment with ASOs gave promising results. ASOs seem to be an ideal strategy for the knock down of HDAC6 in motor neurons as we could use unassisted uptake. This means that we only had to add these ASOs to the culture medium at sufficiently high concentrations which had no toxic effects on the cells⁶. We treated mature motor neurons for 1 week with ASOs directed against HDAC6 and observed a 50% decrease of HDAC6 (revised Figure 6). After knock down of HDAC6, the patient-derived motor neurons showed an increase in mitochondrial transport which is in line with the reported specificity of the HDAC6 inhibitors.

- The supplemental table/data is a bit confusing. A legend, title or key should be added to place this data in context.

Response: We thank the reviewer for pointing out this mistake. Please see the revised version as we added the title and a legend for the supplemental data.

Reviewer#2:

In this study, the authors investigate the pathological mechanisms associated with FUS-related ALS and test potential therapeutic strategies. They generated iPSCs from ALS patients carrying different FUS mutations, healthy controls and differentiated them into motor neurons. For some experiments, they also used iPSC lines in which the mutation was corrected. They found that neurons expressing mutant FUS show cytoplasmic accumulation of FUS, hypo-excitability, synaptic and axonal transport defects, consistent with many previous reports. They further demonstrated that axonal transport defects could be rescued by HDAC6 inhibitors. This manuscript is well written and easy to follow. However, the quality of the figures needs to be improved. For most panels, it is very difficult to read the labels. The main concern about this work is that, although potentially interesting, it does not provide sufficient mechanistic insight. The link between FUS and HDAC6 has already been reported before, and the authors did not investigate in detail the mechanism by which HDAC6 inhibitors suppressed axonal transport defects caused by mutant FUS.

Response: We are grateful for the overall appreciation of our manuscript. Moreover, we apologize for the poor quality of our figures, which is at least partially due to the low quality of the PDF version of our manuscript. We improved our figures and also uploaded them separately at a higher resolution.

The link previously reported between FUS and HDAC6 was situated at the mRNA level⁷. It was discovered that downregulation of FUS (or TDP-43) downregulated the mRNA level of HDAC6. Our current hypothesis is that increased α -tubulin acetylation by inhibition of the deacetylating function of HDAC6 is responsible for the rescue of the axonal transport defects. These effects don't seem to be specific for defects caused by the presence of mutant FUS. We believe that the HDAC6 inhibitors could have a more general effect on the improvement of axonal transport efficiency by increasing the acetylation level of α -tubulin. We have preliminary data that similar mechanisms are present in ALS-iPSC-MNs carrying other mutations (TDP-43 and C9orf72). It is not yet clear how the axonal transport defects are caused by mutant FUS (or mutant TDP-43 or hexanucleotide repeats in C9orf72). One possibility is that axonal transport defects are the result of stress imposed to the cell by mutant FUS (or other mutant genes).

[Redacted]

Specific points:

1. Figure 2 A and B: authors need to improve the quality of the images. The images of iPSCs and neurons are not of sufficient clarity to support the author's conclusions. It looks like FUS is also in the cell body of control neurons?!

Response: We apologize for the quality of our images. We took new images with higher resolution. Please see revised Figure 2A and B.

2. Figure 2 E – K: only one control line was used so it is difficult to draw meaningful conclusions. Other control lines should be used.

Response: The reviewer is right as we indeed had only one of our own control lines available at the moment that these experiments were performed. This problem was solved by using two healthy control lines that were recently published⁸. These experiments were done in parallel and are now incorporated in our manuscript. Please see revised Figure 2E-K.

3. The total number of mitochondria at 6-7 weeks is half of the number at 2-4 weeks (Figure 3 B and D) so there should be a decrease in mitochondria over time even in control neurons. The authors should extend their time-course to 6-7 weeks after plating to clarify this point. The same applies to the number of moving mitochondria (Figure 3 C and D). Why are the mitochondria moving less at the later time point (in both control and patient neurons)?

Response: We agree with the referee that this is an interesting observation. However, we believe that this difference is at least partially due to technical problems. However, these problems didn't influence our conclusions as we always compared patient and control lines in parallel. After a longer time in culture, the motor neurons start to cluster and some of the axons sometimes detach, which introduces more variation in the determination of the number of mitochondria, as well as in the axonal transport measurements. As a consequence, we decided to conduct most of our experiments before the 5th week after plating (when these technical problems were absent). As shown in Figure 3, the differences between patient and control are still there after a longer time in culture and seem even more pronounced which is in line with the cells experiencing more mutant FUS induced stress as a function of time in culture.

4. Quantification of the number of neurons positive for different markers should be performed to show that a more than 3-fold increase in FUS expression does not affect neuronal differentiation capacity.

Response: In response to this question, we quantified the markers Chat and Smi32 at week 4 of differentiation and no differences were observed after overexpression of FUS (Suppl. Figure 4).

5. Images in Figure 5 A-B are not of sufficient resolution/quality to determine co-localization. PDI and TOM20 should not be present in the nucleus of the neuron. It's not clear how this quantification was performed.

Response: In part, this was due to the low quality of the pdf version of our manuscript. Moreover, we took new pictures with a higher resolution. Please see revised Figure 5. In order to quantify these images, we used ImageJ with the Coloc2 plug-in to obtain the Pearson's r correlation value as our read out for the localization. The values ranged from 1 for images in which fluorescence intensities were perfectly colocalizing, to -1 for images in which fluorescence intensities were perfectly, but inversely, correlated to one another. Values near zero reflect distributions of signals that are uncorrelated with one another. We have also incorporated this information in material and methods in our manuscript.

6. How do the authors explain the effect of the HDAC6 inhibitors, as no defects in acetylation were found in the patient neurons? Is the effect specific to HDAC6 activity? Additional experiments, such as CRISPR and RNAi knock-down of HDAC6 are needed to clarify this point. How much inhibition is needed to observe a protective effect?

Response: As explained in the answer to referee 1 on a similar question, we tried both lentiviral shRNA, CRISPR/Cas9 as well as ASOs (antisense oligonucleotides) to knock down HDAC6 in our cultured motor neurons. Only the ASOs gave reliable results. Unassisted uptake by adding these ASOs to the culture medium had no toxic effect on the cells⁶. We treated mature motor neurons for 1 week by using ASOs and observed a 50% decrease of HDAC6 (revised Figure 6). The patient-derived motor neurons showed an increase in mitochondrial transport after knock down of HDAC6. This does not only confirm the rescue effect of the HDAC6 inhibitors but it also proves the specificity of the inhibitors that we have used. Based on these experiments with ASOs, we can conclude that a decrease of HDAC6 by 50% is already sufficient to give a protective effect. How much inhibition is exactly necessary is difficult to say as it was impossible to perform dose-response curves for the different drugs (as explained in the answer to the third question of referee 1).

For the mechanism underlying the positive effect of HDAC6 inhibition, we refer to our answer on the general question. We consider the axonal transport defect as a pathological output which is the result of cell stress caused by mutant FUS (or other mutated gene products). HDAC6 inhibition increases the transport efficiency by increasing the acetylation level of α -tubulin, while the healthy cells already reached the maximal transport efficiency without any effect after HDAC6 inhibition. This hypothesis is illustrated in Figure 6 Overall, we believe that it could be a general rescue method to overcome axonal transport problems in ALS and related diseases.

7. Does inhibition/KD of HDAC6 affect the phosphatidylcholine levels in motor neurons?

Response: This is an interesting suggestion, Please see Suppl. Figure 7. We see a rescue effect by using ACY738 treatment but did not see obvious rescue by silencing HDAC6 in our motor neurons.

8. What are the consequences of the axonal transport defects observed? Do they cause cell death? Can it be modulated by HDAC6 inhibition/KD?

Response: We did not observe obvious cell death in our cultures. However, slightly higher values for apoptotic markers were obtained in our mutant cultures (results not shown). As this was very limited (and extremely variable), it could (unfortunately) not be used as a readout to test the effect of HDAC6 inhibitors. We believe that one of the reasons that differentiated motor neurons are not dying in culture could be that we provide them with a very rich culture medium in order to protect them against different forms of stress. Moreover, if those cells that would be most vulnerable to axonal transport defects would all die, it would be impossible to study the phenotype of these cells (as these cells are gone). Most likely, we will have to impose different forms of stress to the cultured cells in order to induce more pronounced and consistent cell death. We consider this as beyond the scope of our current study also because there is already a lot of evidence that axonal transport defects play an important role in ALS⁹. Their conclusion is that: "The link between axonal transport defects and ALS and other neurodegenerative diseases is very strong". We also fully agree with their statement that: "It seems most likely that restoring axonal transport will not be a magic bullet treatment but will be of benefit in combination with other treatments targeting the diverse cellular mechanism associated with disease such as protein aggregation and mitochondrial damage. Indeed, the widespread and early nature of axonal transport defects in ALS suggests that these defects will have to be addressed if treatment is to be effective". The ambition of our paper is to contribute to the

development of these new therapeutic strategies that could reverse these axonal transport defects using motor neurons obtained from iPSCs.

[Redacted]

Minor points:

- Table S1: what is the age at biopsy of patient 2/1?

Response: The biopsy of patient 2/1 was taken when the patient was 38 years old. As this presymptomatic patient did not want to know whether or not the gene was mutated, we deliberately omitted this information from our paper as it could otherwise reveal the identity of patient 2/1 in case the patient would see the manuscript. This is in line with the recommendations of our ethical committee.

- Figure S2B: which genes were analyzed by qPCR?

Response: This analysis was done using a commercial kit (Life Technologies - A15870 - HPSC scorecard panel 384). The qPCR was done using the specific primer panel provided by the company and analyzed using the scores for the marker of each germ layers. The identity of the markers was not revealed by the company. We have included this information in the manuscript. these markers are:

Ectoderm: CDH9, COL2A1, DMBX1, DRD4, EN1, LMX1A, MAP2, MYO3B, NOS2, NR2F1/NR2F2, NR2F2, OLFM3, PAPLN, PAX3, PAX6, POU4F1, PRKCA, SDC2, SOX1, TRPM8, WNT1, ZBTB16.

Endoderm: AFP, CABP7, CDH20, CLDN1, CPLX2, ELAVL3, EOMES, FOXA1, FOXA2, FOXP2, GATA4, GATA6, HHEX, HMP19, HNF1B, HNF4A, KLF5, LEFTY1, LEFTY2, NODAL, PHOX2B, POU3F3, PRDM1, RXRG, SOX17, SST.

Mesoderm: FGF4, GDF3, NPPB, NR5A2, PTHLHT, ABCA4, ALOX15, BMP10, CDH5, CDX2, COLEC10, ESM1, FCN3, FOXF1, HAND1, HAND2, HEY1, HOPX, IL6ST, NKX2-5, ODAM, PDGFRA, PLVAP, RGS4, SNAI2, TBX3, TM4SF1.

Self-renew: CXCL5, DNMT3B, HESX1, IDO1, LCK, NANOG, POU5F1, SOX2, TRIM22

- on page 11: “Fig. 5C and D” should be “5D and E”.

Response: We thank the reviewer for pointing out this mistake and we corrected it in our revised version.

Reviewer#3:

Guo et al. have investigated the relationship between FUS mutations and axonal transport defects in iPSC-derived motor neurons carrying ALS-related FUS mutations. They also observe that HDAC6 inhibition restores a normal axonal transport in mutant FUS iPSC-derived motor neurons and propose that HDAC6 inhibitors may represent a new therapeutic strategy in ALS linked to FUS mutations. The authors have generated iPSC cells from ALS patients (R521H, P525L) and control

fibroblasts and used CRISPR/Cas9 method to correct one line (R521H) into isogenic line carrying R521R.

They determined that R521H and P525L- FUS proteins are mislocalized in the cytoplasm of fibroblasts, undifferentiated and four weeks differentiated iPS-derived motor neurons. They also show that mutant FUS motor neurons exhibit hypo-excitability, which is consistent with a previous report (Naujock et al., 2016). They propose that mutant FUS motor neurons present progressive axonal transport defects by studying the mobility of mitochondria and endoplasmic reticulum. These results are consistent with a previous report of axonal transport defect in *Drosophila* carrying FUS mutations (Baldwin et al., 2016). By using corrected isogenic line R521R, the authors established the causative link between FUS mutations and the axonal transport defect. In addition, they show a decrease of mitochondria-associated ER membranes (MAM) in neurites of mutant FUS motor neurons. MAMs are also involved in the regulation of the phospholipid metabolism and reduced phosphatidylcholine (PC) levels were found in culture medium from mutant motor neurons.

Finally, the authors have tested the potential of HDAC6 in reversing abnormal phenotype in FUS mutant iPS-derived motor neurons. Indeed, they had previously reported that abnormal axonal transport in a mouse model expressing mutant HSPB1-induced Charcot-Marie-Tooth disease (CMT2) could be restored by inhibition of HDAC6 (d'Ydewalle, C. et al., 2011) through an increase of α -tubulin acetylating activity. Two years later, Taes et al., have also shown that Hdac6 deletion delays disease progression in the SOD1G93A mouse model of ALS¹⁰ (Taes et al., 2013). While no defect of α -tubulin acetylation was detected in mutant FUS motor neurons, the authors determined that treatment by HDAC6 inhibitors such as tubastatin A and ACY738 restored both MAM and axonal transport defects. ACY738 is already in clinical trials for cancer and has a blood-brain-barrier permeability (Falkenberg et al., 2014; Majid et al., 2015), opening the opportunity for development of a new therapeutic strategy in ALS.

The link between FUS mutations and axonal transport was already established (Baldwin et al., 2016), and the novelty comes mostly from the system used: iPS-derived motor neurons and isogenic corrected cell lines. While HDAC6 inhibitor was already tested in CMT2 and the deletion of Hdac6 was promising in ALS-SOD1 mouse model, HDAC6 inhibitors were never tested in ALS-FUS model. Hence the most important finding of this study is the effect of HDAC6 inhibitors in rescuing transport defect associated to ALS-FUS mutations in less than 24 hours.

Overall, this study uses innovative tools (iPS-derived motor neurons; CRISPR edited control line and inducible ES lines overexpressing mutant and wild-type) and confirms a role for transport deficit in FUS-ALS disease. Importantly, the authors identify a potential new therapeutic target in FUS-ALS. In this reviewer's view the tools and findings from this study are of high relevance for the field, however the quality of the evidence is often difficult to appreciate and several concerns should be addressed to reach the level required for publication in Nature Communication.

Response: We thank the reviewer for reading our article in detail and for this nice and constructive summary.

Majors concerns:

1. A general concern is the presentation of the data. Indeed, Figures and fonts are extremely small and often not readable on a printed version. Most importantly images provided on Figures 1F, 2A-B, 3A, 3F, 5A, and Supplementary Figures S2C and S8A are not appropriate (low magnification; deem staining) and results difficult to evaluate.

Response: We apologize for the poor quality of our images which is at least partially due to the conversion of our manuscript into a PDF. Please see the revised version (in separate files) of Figures.

2. The authors show that axonal transport defects are rescued by only 24h of treatment with HDAC6 inhibitors. It would be important to demonstrate that HDAC6 inhibitors also rescue other phenotypes identified in FUS motor neurons. Indeed, the authors should perform electrophysiological experiment on FUS motor neurons treated with Tubastatin A or ACY-738 in order to assess whether HDAC6 inhibitors can rescue the hypo-excitability of FUS motor neurons. They should also test whether Tubastatin A or ACY-738 treatment restore the level of phosphatidylcholine in FUS motor neurons.

Response: We thank the reviewer for these interesting suggestions. Although axonal transport is the main focus of this article, we tried to extend the effect of HDAC6 inhibition on the other phenotypes. Due to some technical problems, it was unfortunately not possible to perform this for the electrophysiological experiments during the past three months. However, we did not observe changes in the FUS mislocalization after we treated our patient cells with HDAC6 inhibitors (See Suppl Figure 8). Moreover, we observed a slight rescue for the PC level after we added HDAC6 inhibitors to motor neurons (See Suppl Figure 7B).

3. The authors should discuss whether HDAC6 inhibition is associated with toxicity and whether treatment longer than 24h is tolerated by FUS mutant motor neurons in culture.

Response: We fully agree with the referee that the toxicity of the compounds used in our study needs to be addressed before considering it as potential drugs that can be used in a clinical context. This is a very similar question as the one of reviewer 1 (second question). We refer to this answer for details concerning the potential toxicity of the HDAC6 inhibitors. In preclinical studies, tubastatin A is a widely used compound which showed no toxicity at the concentrations used. A similar remark can be made for ACY-738.

4. It is also not clear whether the authors have identified anterograde or retrograde transport deficits. Most importantly, while they propose that FUS motor neurons have an axonal transport deficit their analysis does not seem to selectively monitor axons.

Response: This is indeed a good suggestion. However, due to the fact that the motor neurons in our culture system form many neurites (that intercross) it is in many cases impossible to be 100% sure to which cell body the neurite belongs. We tried to distinguish the direction by seeding the motor neurons at very low densities. Even then it was often very difficult to tell the difference between retrograde and anterograde transport in these extremely long neurites. Moreover, the variation became very high when we tried to separate retrograde and anterograde transport in those cases for which we could determine the direction. We also agree with the referee that we don't make a difference between axons and dendrites in our cultures (we even don't know whether there is a difference). Strictly spoken, neurite would be a better term. However, we prefer to call what we measure 'axonal transport' as there is no good alternative and as we believe that these measurements in the longest and most prominent neurite is representative for what happens in the axons.

5. In Figure 3, the transport of mitochondria was measured in control and FUS mutant motor neurons. There is inconsistency between Figure 3E - with 5 moving mitochondria per 100um neurite length after 2, 3 or 4 weeks of differentiation - and Figure 3C with 2 moving

mitochondria per 100um neurite length at 6-7 weeks of differentiation. Is there a progressive reduction of mitochondrial transport in control motor neurons between 4 to 6 weeks of differentiation?

Response: This is a similar question as the one from referee 2 (specific point 3). We indeed observed that the number of mitochondria as well as the transport decreased as a function of the time in culture. As already mentioned, we believe that this could also have a technical cause. As a consequence, we decided to conduct most of the experiments before the 5th weeks after plating and always in parallel. As is shown in Figure 3, the differences between patient and controls were still there and are even more pronounced at a later time point.

6. In Figure 1F and S2C it is almost impossible to see the Dapi staining and to appreciate the proportion of positive motor neurons.

Response: Please see the revised version of Figure 1F and Suppl Figure 2C.

7. Figure 2A shows increased cytoplasmic staining in mutant fibroblasts and motor neurons but not “nuclear clearance” as claimed on page 6.

Response: Nuclear clearance also exists in fibroblasts and we marked those cells showing this ‘phenotype’ with a white arrow (Figure 2A).

8. It is not clear why only one control line was included in the electrophysiology studies on Figure 2. Considering the important variability of the results between lines it seems important not to rely only on 1 control.

Response: This is a similar comment as the one from referee 2 (specific point 2). Please see revised Figure 2E-K in attached files. We added two healthy controls (published lines⁸). The experiments were done in parallel.

9. The result on Fig 2K should be clarified. It seems that there is a reduction of PSCs amplitudes except for 1 mutant line. Also the error bar on the patient “pooled clones” is surprisingly small considering the variability between individual clones.

Response: As indicated in the figure legend below, we used SEMs for the error bar shown. Since the patch clamp data contain 50 to 100 cells per condition, the error bars became indeed very small after calculating the SEMs. Please see the data with SD error bars and SEM error bars below.

10. The quality/size of Figure 3F does not allow the readers to appreciate mitochondrial integrity. Similarly, the identification of mitochondria-associated ER membranes (MAMs) by

co-staining of Tom20 and PDI can not be appreciated on Fig 5A (the colocalization is not clear even in the small zoom views).

Response: We apologize for the poor quality of this figure. Please see the revised version of Figure 3F and Fig 5A.

11. The mutant FUS motor neurons do not present an increase of alpha-tubulin acetylation, while HDAC6 inhibitors restore mitochondrial and ER transport defect. The authors should better discuss the mechanism of action of HDAC6 inhibitors on axonal transport defect. The rationale to test HDAC6 inhibitors should also be included in the introduction.

Response: The constructive suggestion is in line with one of the comments of the second referee. We believe that the HDAC6 inhibition rescues axonal transport defects because of the increased acetylation level of α -tubulin. For a detailed answer, we refer to the answer to the general question of referee 2.

We have added more information for the reasoning behind the choice of HDAC6 inhibitors in the introduction of our manuscript.

12. The authors should discuss previous reports that TDP43 and FUS bind and regulate the level of HDAC6 mRNA (Kim JBC 2010; Fiesel EMBO 2010; Fiesel Mol Neurod 2011; Tollervey Nat neurosc 2011; Polymenidou Nat Neurosc 2011; Miskiewicz cell Rep 2014; Odagiri BBRs 2013).

Response: We thank the reviewer for these suggestions. Please see the new and marked paragraph in the discussion part of our manuscript.

13. The authors should better discuss the results indicating that there might be an inter-regulation between AMPA receptor and FUS (Supplemental Fig 3G, H).

Response: We do not really understand the point the reviewer wants to make as we have no evidence that mutant FUS influences the AMPA receptor.

14. The authors propose that the phenotypes observed are due to a gain of toxic function. However, to make this claim they would need to knock-down FUS in iPS-derived motor neurons and measure transport and excitability properties.

Response: We used ASOs to successfully knock down FUS in one isogenic control line and did not see any change in the axonal transport which is in line with our 'gain of function' hypothesis in relation to the axonal transport phenotype. We have mentioned this in our manuscript. Within the time frame of this revision, it was not possible to do similar patch clamp experiments. We don't consider this as a major problem as the focus (and novelty) of our manuscript is based on the axonal transport phenotype and the rescue of this phenotype by selective HDAC6 inhibitors.

Minor concerns:

1. It would be interesting to show the cellular localization of FUS in the isogenic cell line R21R and in hESC lines overexpressing mutant and wild-type FUS.

Response: We do indeed see a correction of the mislocalization in our isogenic control and we also observed FUS mislocalization after we overexpressed mutant FUS in hESC. Please see revised Suppl Figure 4.

2. The following references should be included in the manuscript:
- Taes et al. HMG (2013). Hdac6 deletion delays disease progression in the SOD1G93A mouse model of ALS.
 - Wang et al. Mol Ther (2017). Motor-Coordination and Cognitive Dysfunction Caused by Mutant TDP-43 Could Be Reversed by Inhibiting Its Mitochondrial Localization.
 - Vance et al. Science (2009) should be included along with Kwiatkowski et al. Science (2009).
 - Appropriate references should be included to the following sentence on page 3: “ in iPSC models, cytoplasmic mislocalization of mutant FUS was reported by three independent groups in their patient-derived motor neurons.”

Response: We added these references to our manuscript.

3. On page 23, Reference 12 lacks the year of publication: Devlin et al. Human iPSC-derived motoneurons harbouring TARDBP or C9ORF72 ALS mutations are dysfunctional despite maintaining viability. Nat. Commun. 6, 1–12 (1AD).

Response: We thank the reviewer for pointing out this mistake and we corrected this reference.

4. On page 5, the sentence “the origin of the iPSCs was confirmed by SNP analysis (Suppl Table 2)” should be removed since Suppl Table 2 does not correspond to SNP analysis. It is not clear if the authors want to include or not this analysis (provided at the end without a table number and referred as “data not shown” in the text).

Response: We apologize for this confusion. Please see the revised version of the Suppl. Table 2.

5. On page 8, “Fig 3G” should be removed from the sentence “change in mitochondria in the soma or axons of patient-derived MNs compared to controls (Fig. 3F, 3G)”, and added to the sentence “We also measured axonal transport of ER vesicles using ER-Tracker (Fig. 3H, 3I).”

Response: We corrected this mistake. Please see the revised version for the Figure3.

References:

1. Benoy, V., Berghe, P. Vanden, Jarpe, M. & Damme, P. Van. Development of Improved HDAC6 Inhibitors as Pharmacological Therapy for Axonal Charcot – Marie – Tooth Disease. *Neurotherapeutics* **14**, 417–428 (2017).
2. d’Ydewalle, C. *et al.* HDAC6 inhibitors reverse axonal loss in a mouse model of mutant HSPB1-induced Charcot-Marie-Tooth disease. *Nat. Med.* **17**, 968–974 (2011).
3. Butler, K. V *et al.* Rational Design and Simple Chemistry Yield a Superior , Neuroprotective HDAC6 Inhibitor , Tubastatin A. 10842–10846 (2010).
4. Xu, X., Alan, P. & Pozzo-miller, L. A selective histone deacetylase-6 inhibitor improves BDNF trafficking in hippocampal neurons from Mecp2 knockout mice : implications for Rett syndrome. **8**, 1–9 (2014).
5. Zhang, L. *et al.* Tubastatin A/ACY-1215 Improves Cognition in Alzheimer’s Disease Transgenic Mice. *J. Alzheimer’s Dis.* **41**, 1193–1205 (2014).
6. Stein, C. A. *et al.* Efficient gene silencing by delivery of locked nucleic acid antisense

oligonucleotides , unassisted by transfection reagents. **38**, 1–8 (2010).

7. Kim, S. H., Shanware, N. P., Bowler, M. J. & Tibbetts, R. S. Amyotrophic lateral sclerosis-associated proteins TDP-43 and FUS/TLS function in a common biochemical complex to co-regulate HDAC6 mRNA. *J. Biol. Chem.* **285**, 34097–34105 (2010).
8. Naujock, M. *et al.* 4-Aminopyridine Induced Activity Rescues Hypoexcitable Motor Neurons from Amyotrophic Lateral Sclerosis Patient-Derived Induced. *Stem Cells* **34**, 1563–1575 (2016).
9. De Vos, K. J. & Hafezparast, M. Neurobiology of axonal transport defects in motor neuron diseases: Opportunities for translational research? *Neurobiol. Dis.* (2017). doi:10.1016/j.nbd.2017.02.004
10. Taes, I. *et al.* Hdac6 deletion delays disease progression in the SOD1 G93A mouse model of. *Hum. Mol. Genet.* 1–23 (2013).

REVIEWERS' COMMENTS:

Reviewer #1 (Remarks to the Author):

The authors have responded appropriately to the concerns that I and the other reviewers raised. The manuscript is improved in quality and substance.

Reviewer #2 (Remarks to the Author):

I don't have any further questions and recommend the acceptance of this manuscript.

Reviewer #3 (Remarks to the Author):

In this revised version Guo et al. have largely addressed my concerns. In particular they have included important new experiments showing that FUS loss of function is not responsible for the phenotype and that genetic silencing of HDAC6 had similar effects as pharmacological inhibition. The presentation of the results has also been improved. I believe that this work will be of high interest for the field and recommend its publication in Nature Communication.

Minor concerns:

- The quality of the images is overall improved. However it is still difficult to read the captions in Fig 1F, or to see the localization of FUS in iPS (Fig 2A, right panel). Also, the scheme on Fig 4 is too small and can not be read.
- It seems that there is a mistake in Suppl Fig 9 where cell body and dendrites should be shown for both mutant and control. The upper left images labeled ctr1 appear to be mutant when compared with Fig 5A.
- The axis on Fig 3H should be "Total ER/100um" instead of "total mito/100um".
- There are several typos in the sentence "Only if we overexpress...." on page 10.
- Table S2 is still missing a title and legend.

ANSWERS TO THE REVIEWERS

Reviewer #1 (Remarks to the Author):

The authors have responded appropriately to the concerns that I and the other reviewers raised. The manuscript is improved in quality and substance.

Response: We thank for the very positive response from this reviewer.

Reviewer #2 (Remarks to the Author):

I don't have any further questions and recommend the acceptance of this manuscript.

Response: We thank for the very positive response from this reviewer.

Reviewer #3 (Remarks to the Author):

In this revised version Guo et al. have largely addressed my concerns. In particular they have included important new experiments showing that FUS loss of function is not responsible for the phenotype and that genetic silencing of HDAC6 had similar effects as pharmacological inhibition. The presentation of the results has also been improved. I believe that this work will be of high interest for the field and recommend its publication in Nature Communication.

Response: We thank for the general positive response from this reviewer.

Minor concerns:

- The quality of the images is overall improved. However it is still difficult to read the captions in Fig 1F, or to see the localization of FUS in iPSC (Fig 2A, right panel). Also, the scheme on Fig 4 is too small and can not be read.

Response: We thank this reviewer for the detailed checking of our manuscript. We changed the captions and have placed it above the panel (F) in Fig1 and we included a confocal image with a higher magnification to show FUS in the iPSC (Fig 2A, right panel). We also made a new version of the scheme containing characters with a larger font size.

- It seems that there is a mistake in Suppl Fig 9 where cell body and dendrites should be shown for both mutant and control. The upper left images labeled ctr1 appear to be mutant when compared with Fig 5A.

Response: We are sorry for this mistake. We have added control lines in Suppl Fig9.

- The axis on Fig 3H should be "Total ER/100um" instead of "total mito/100um".

Response: We are sorry for this mistake. We have changed according to the suggestion of the referee.

- There are several typos in the sentence "Only if we overexpress...." on page 10.

Response: We have corrected all the typos.

- Table S2 is still missing a title and legend.

Response: We have added the title.